# Informed by Cancer Stem Cells of Solid Tumors: Advances in Treatments Targeting Tumor-Promoting Factors and Pathways

**DOI:** 10.3390/ijms25074102

**Published:** 2024-04-07

**Authors:** Maya R. MacLean, Olivia L. Walker, Raj Pranap Arun, Wasundara Fernando, Paola Marcato

**Affiliations:** 1Department of Pathology, Dalhousie University, Halifax, NS B3H 4R2, Canada; maya.maclean@dal.ca (M.R.M.); olivia.walker@dal.ca (O.L.W.); rpranaparun@dal.ca (R.P.A.); wasufer@dal.ca (W.F.); 2Department of Biology, Acadia University, Wolfville, NS B4P 2R6, Canada; 3Department of Microbiology and Immunology, Dalhousie University, Halifax, NS B3H 4R2, Canada; 4Nova Scotia Health Authority, Halifax, NS B3H 4R2, Canada

**Keywords:** cancer stem cells, therapeutic, target, marker, clinical trial

## Abstract

Cancer stem cells (CSCs) represent a subpopulation within tumors that promote cancer progression, metastasis, and recurrence due to their self-renewal capacity and resistance to conventional therapies. CSC-specific markers and signaling pathways highly active in CSCs have emerged as a promising strategy for improving patient outcomes. This review provides a comprehensive overview of the therapeutic targets associated with CSCs of solid tumors across various cancer types, including key molecular markers aldehyde dehydrogenases, CD44, epithelial cellular adhesion molecule, and CD133 and signaling pathways such as Wnt/β-catenin, Notch, and Sonic Hedgehog. We discuss a wide array of therapeutic modalities ranging from targeted antibodies, small molecule inhibitors, and near-infrared photoimmunotherapy to advanced genetic approaches like RNA interference, CRISPR/Cas9 technology, aptamers, antisense oligonucleotides, chimeric antigen receptor (CAR) T cells, CAR natural killer cells, bispecific T cell engagers, immunotoxins, drug-antibody conjugates, therapeutic peptides, and dendritic cell vaccines. This review spans developments from preclinical investigations to ongoing clinical trials, highlighting the innovative targeting strategies that have been informed by CSC-associated pathways and molecules to overcome therapeutic resistance. We aim to provide insights into the potential of these therapies to revolutionize cancer treatment, underscoring the critical need for a multi-faceted approach in the battle against cancer. This comprehensive analysis demonstrates how advances made in the CSC field have informed significant developments in novel targeted therapeutic approaches, with the ultimate goal of achieving more effective and durable responses in cancer patients.

## 1. Introduction

In the last twenty years, cancer recurrence, metastasis, and chemoresistance have increasingly been associated with cancer stem cells (CSC). CSCs are a subpopulation of cancer cells defined by their proclivity for self-renewal and differentiation, as well as tumorigenicity [1,2,3]. Indeed, an isolated population of CSCs could give rise to new tumors, while simultaneously repopulating the tumor with both CSC and non-CSC populations [1]. CSC populations can be defined by different cell surface and intracellular markers [4]. Interestingly, despite their role as aggressive tumor initiators, CSCs only comprise a small percentage of bulk tumor volume, with reports suggesting ranges from 0.02% to 25% [5,6]. The importance of CSCs in the development of chemoresistance has been demonstrated in cancers including breast, head and neck, mesothelioma, gastric, and ovarian cancers and have been extensively reviewed elsewhere [7,8,9,10,11]. Furthermore, CSCs have also been implicated as important contributors to metastasis across multiple cancer types [12,13,14,15,16,17,18]. Many current therapies that are designed to target fast proliferating bulk tumor cells are much less effective at clearing the highly aggressive and possibly quiescent CSC populations, which can ultimately lead to metastasis and recurrence [19,20,21]. Thus, developing therapies that can target aggressive CSC populations has been of increasing interest in the last 20 years.

There are several CSC markers and associated pathways thar are involved in maintaining stemness and the aggressive CSC phenotype across multiple cancers and are potential candidates as anti-cancer targets. In this review, we discuss the status of the therapeutics available that target CSC markers and pathways more active in CSCs across multiple cancer types, focusing on solid tumors. We include discussion of advances in targeting aldehyde dehydrogenases (ALDHs), epithelial cell adhesion molecule (EpCAM), Cluster differentiation 44 (CD44), CD55, C-X-C receptor 4 (CXCR4), CD133, Nanog, neurogenic locus notch homolog protein (Notch), Wnt/β-catenin, Sonic hedgehog (SHH), and Sry-box 2 (SOX2). Figure 1 represents the cellular location of the CSC markers and factors/pathways associated with CSCs that are discussed in this review.

It is notable that not all the molecules included in this review are necessarily CSC markers (e.g., CXCR4), but we have included them because they are enriched in CSCs or promote CSC aggressiveness/maintenance, and as such, targeting them could reduce CSCs. Furthermore, the properties of tumor progression and chemoresistance that have been ascribed to these molecules are also intrinsically associated with CSCs and cannot be separated from CSCs. In the reviewed studies, drugs that block tumor progression and chemoresistance are also described to inhibit CSC populations. Furthermore, in our review of drug tumor studies, not all studies specifically addressed effects on the CSC population; however, for completeness, we included those studies. In general, the molecules that are included in this review are drug targets outside of their association with CSCs because they promote tumor progression and chemoresistance. Finally, we use the term CSC, which is often equated to tumor-initiating cell (TIC); however, TICs are not necessarily CSCs and vice-versa. TIC is a functional term defined by the ability to form new tumors in a suitable host organism. CSCs may not necessarily have this ability. CSCs display stemness properties, which TICs may not have.

## 2. Aldehyde Dehydrogenases

ALDHs are a group of nicotinamide adenine dinucleotide (NAD^+^) or NAD phosphate (NAD^+^) dependent enzymes that catalyze the irreversible oxidation of aldehydes to carboxylic acids. The ALDH superfamily is composed of 19 different isoforms, which exhibit distinct tissue expression profiles and subcellular location, as well as substrate specificity and function. In general, ALDHs serve as important regulators of cellular homeostasis, removing toxic aldehydes that are produced during metabolic processes, such as those arising from amino acid catabolism, lipid peroxidation, and exogenous xenobiotics [22]. Additionally, there are several isoforms (i.e., ALDH1A1, ALDH1A2, ALDH1A3) that are responsible for converting the vitamin A metabolite, retinaldehyde, to retinoic acid (RA), and have been associated with CSCs [23,24]. Retinoic acid can modulate the expression of genes, including those responsible for apoptosis, cellular proliferation, and differentiation [25,26].

Due to their role in supporting cellular homeostasis and metabolism, it is unsurprising that there are several ALDH isoforms that have been implicated in cancer progression as CSC markers. ALDH activity in cancer stem cells is most commonly detected by the Aldefluor assay, which measures the conversion of BODIPY aminoacetaldehyde to fluorescent reaction product BODIPY aminoacetate [23]. High ALDH activity can be attributed to several ALDH isoforms, including ALDH1A1, ALDH1A2, ALDH1A3, ALDH2, and ALDH3A1 [27,28,29,30]. It is well-appreciated that these isoforms are involved in maintaining the stemness of CSCs across multiple cancer types, including breast, prostate, colorectal, pancreatic, melanoma, lung, liver, and brain [30,31,32,33,34,35,36]. Indeed, ALDH isoforms have been linked to enhanced clonogenicity and tumorigenicity, as well as chemoresistance and metastasis [13,30,31,33,37]. Recently, there has been increasing interest in the development of ALDH inhibitors to treat cancer and target aggressive CSC populations. The following subsections discuss each of the ALDH isoforms that have been linked to CSCs followed by the development of inhibitors that target these isoforms specifically. We also include a subsection on inhibitors that have pan-ALDH inhibitory activity. Table 1 provides a summary of the inhibitors.

### 2.1. ALDH1A1

ALDH1A1 has been linked to stemness features such as chemotherapy resistance and worse prognosis in glioma, prostate, esophageal, gastric, ovarian, and breast cancer [30,35,38,39,40,41]. Several ALDH1A1-specific inhibitors have been developed, including NCT-501, CM37, and 974. In head and neck squamous cell carcinoma (HNSCC), ALDH1A1 was enriched in cisplatin and 5-fluorouacil-resistant squamous cell carcinoma (SCC) Cal-27 cell lines, which displayed enhanced stemness features such as spheroid formation, clonogenicity, and migratory potential [42]. Treatment with 40 nM and 80 nM of NCT-501, a theophylline-based reversible inhibitor of ALDH1A1, significantly reduced Cal-27 spheroid size and formation, as well as migratory capacity [42,43]. Furthermore, intra-tumoral injection of 100 μg of NCT-501 in Cal-27 cisplatin-resistant xenografts in mice inhibited tumor growth by 78% compared to control and induced sensitivity to cisplatin in patient-derived explant samples [42]. However, NCT-501 exhibited low cellular activity in pancreatic and colorectal cancer cell lines, MIA PaCa-2 and HT-29, with an IC50 > 4 μM, indicating that its effect may be cell-line specific [44]. CM37, another small molecule ALDH1A1 inhibitor, reduced spheroid formation in ovarian cancer cell lines OVCAR8 and OVCAR3 at 5 and 20 μM, respectively; however, an IC50 has yet to be determined [45]. In another recent study, Compound 974 demonstrated an IC50 of 470 nM for binding with purified ALDH1A1 [46]. Pre-treatment of OVCAR3 cells with 5 μM of Compound 974 inhibited tumor formation in vivo in a limiting dilution analysis, indicating a reduction in CSC frequency [46]. Furthermore, Compound 974 inhibited chemotherapy-induced senescence in OVCAR3 cells [46].

### 2.2. ALDH1A3

It is well appreciated that ALDH1A3 is an important driver of stemness in multiple cancer types, including breast, melanoma, glioblastoma, prostate, non-small cell lung cancer (NSCLC), head and neck cancer, and colon cancer [47]. Indeed, ALDH1A3 has been shown to enhance metastasis, clonogenicity, and chemoresistance [48,49]. There are a few compounds that show specificity toward ALDH1A3, including MF-7, GA11, MCI-INI-3, NR6, and YD1701 [12,50,51,52,53]. Intraperitoneal injections of GA11 (20 mg/kg) attenuated the growth of mesenchymal glioma tumor-derived MES83 and MES267 cells in mice [50]. Later, modifications of GA11 to MF-7 increased its IC50 in cell-free assays from 4.7 μM to 22.8 μM; however, MF-7 displayed increased functional efficacy compared to GA11 when tested for anti-proliferative activity in the triple-negative breast cancer (TNBC) cell line MDA-MB-468 [12]. While intraperitoneal injections of MF-7 reduced formation and size of MDA-MB-231 and MDA-MB-468 brain metastases, there was a paradoxical increase in the number of lung and bone metastases [12]. NR6, an imidazo [1-2-*a*] pyridine derivative, reduced cell growth and the invasiveness of U87MG glioblastoma and HCT116 colorectal cancer (CRC) cell lines [51]. Recently there have been improvements in the design of other ALDH1A3-specific inhibitors that are able to prevent pocket binding of the ALDH1A3 substrate, retinal. MCI-INI-3 is one such compound designed in silico that reduces RA production in the glioma stem cell line, U87MG, at 15 μM [52]. However, MCI-INI-3 displays cytotoxic effects at 15 μM on non-ALDH1A3-expressing glioma stem cell lines, indicating that it may not have an ALDH1A3-specific impact on glioma stem cell viability [52]. In silico analyses have also revealed YD1701 as a novel ALDH1A3 inhibitor with a reported IC50 of 12.0824 μg/mL [53]. Indeed, YD1701 reduced the invasive capacity of colorectal cancer cell lines SW480, HT29, HCT116, and CRC1 at 0.04 μg/mL and prolonged survival in HCT116-tumor-bearing mice [53].

### 2.3. ALDH2

ALDH2 has been identified as a CSC marker in liver and lung cancers [34,54]. Compared to the other ALDH isoforms, relatively few reports have been published on ALDH2 inhibitors, and fewer still have been tested as anti-cancer agents. CVT-10216 is a selective, reversible inhibitor of ALDH2 (IC50 = 0.029 μM), which was initially developed as a method for treating alcoholism [55]. Similarly, daidzin, an isoflavone derived from the kudzu vine, has an IC50 of 80 nM and has been shown to reduce alcohol consumption in animal models [56,57]. In transwell migration assays, 25 μM of CVT-10216 reduced the migratory capacity of colorectal cancer cell lines HCT-116 and DLD-1 cell lines [13]. Similarly, 100 μM or 25 μM of daidzin reduced DLD-1 and HCT116 migration in transwell migration assays, respectively [13]. Furthermore, both inhibitors reduced the clonogenicity of the HCT-116 and DLD-1 cell lines; however, CVT-10216 exhibited increased efficacy as it reduced clonogenicity at 25 μM in HCT-116 cells, compared to 100 μM with daidzin [13].

### 2.4. ALDH3A1

It is well appreciated that ALDH3A1 is linked to chemotherapy resistance, particularly for agents such as cyclophosphamide, paclitaxel, and doxorubicin [58,59]. ALDH3A1 inhibitors such as dyclonine, CB7, and CB29 have all been shown to enhance cancer cell sensitivity to sulfasalazine and cyclophosphamide, respectively [60,61,62]. Indeed, 50 μM of dyclonine, an oral anesthetic, was shown to inhibit ALDH activity in HSC-4 cells and enhanced sulfasalazine cytotoxicity in ALDH3A1^+^ gastric tumor cells isolated from K19-Wnt1/C2mE mice [60]. Others have reported an IC50 value of 76 μM for dyclonine against ALDH3A1 activity, and while dyclonine has also been reported to inhibit ALDH2, sulfasalazine resistance is attributed to ALDH3A1 expression, not ALDH2 [60,63]. CB57, another ALDH3A1 inhibitor, was shown to be highly specific for ALDH3A1, with an IC50 of 0.2 μM, and did not display any inhibitory effects on any of the ALDH1 or ALDH2 isoforms [61]. Indeed, CB57 sensitized lung and glioblastoma cell lines, A549 and SF767, to mafosamide treatment, evidenced by a significant reduction in cell proliferation when compared to treatment with mafosamide alone [61]. Similarly, CB29, another small molecule inhibitor of ALDH3A1, displayed an IC50 of 16 μM, and decreased the ED50 of mafosamide 1.6-fold in SF767 cells [62]. Using chemoproteomics-enabled covalent ligand screening, Counihan et al. identified EN40, a selective ALDH3A1 inhibitor with an IC50 of 2 μM [64]. Indeed, 10 μM of EN40 impaired survival in lung cancer cell line A549 and significantly reduced tumor growth in A549 at a daily dose of 50 mg/kg [64].

### 2.5. Pan-ALDH Inhibitors

As the 19 ALDH isoforms share up to 40% sequence homology, there are several non-specific ALDH or pan-ALDH inhibitors [22,65]. N,N diethylaminobenzaldehyde (DEAB) is a well-known ALDH inhibitor and has been shown to inhibit ALDH1A1 (IC50 = 0.057 μM), ALDH1A2 (IC50 = 1.2 μM), ALDH1A3 (IC50 = 3 μM), ALDH1B1 (IC50 = 1.2 μM), ALDH2 (IC50 = 0.16 μM), and ALDH5A1 (IC50 = 13 μM) [66]. In melanoma, 200 mg/kg of DEAB has been shown to reduce melanoma xenograft growth and the number of residual melanoma cells [67]. Similarly, DEAB inhibited colony-forming ability in pancreatic cancer cell lines BxPC3 and Panc1 [68]. An in silico screen identified another multi-isoform inhibitor, KS100, which inhibits ALDH1A1 (IC50 = 207 nmol/L), ALDH2 (IC50 = 1410 nmol/L), and ALDH3A1 (IC50 = 240 nmol/L) [69]. Nanoliposomal encapsulation of KS100 (hereafter referred to as NanoKS100) inhibited UACC 903 and 1205-Lu derived melanoma xenograft tumor growth at a dose of 20 mg/kg [69]. Dimethyl ampal thiolester (DIMATE) is an irreversible inhibitor of ALDHs 1 and 3 [70]. DIMATE exhibited cytotoxic effects on acute myeloid leukemia (AML) cell lines, with IC50s ranging between 1.67 μmol/L to 12.2 μmol/L and, interestingly, had no cytotoxic effects on healthy hematopoietic stem cells [71]. Furthermore, 14 and 28 mg/kg of DIMATE significantly reduced the numbers of circulating xenografted AML cells in immunodeficient mice [71]. Another pan-ALDH1 inhibitor, 637A, inhibits ALDH1A1 (IC50 = 246 nM), ALDH1A2 (230 nM), and ALDH1A3 (348 nM) induced calcium-dependent necroptosis in ovarian cancer cell line A2780 [72]. Interestingly, 637A also improves response to cisplatin in SKOV3 cells [72]. Citral has also been shown to inhibit several ALDH isoforms, including ALDH1A3 and ALDH2 [73]. In SKBR3 cells, 10 μM of citral significantly reduced ALDH2-mediated Aldefluor fluorescence, while only 1 μM was required to reduce ALDH1A3-mediated Aldefluor activity in MDA-MB-231 cells [73]. Furthermore, 0.4 mg/kg of nanoparticle encapsulated citral reduced MDA-MB-231-derived tumor growth in mice [73]. Interestingly, disulfiram, a well-known treatment for alcohol abuse, has recently been suggested as a repurposed anti-cancer drug due to its inhibition of ALDH isoforms such as ALDH1A1, ALDH1A3, and ALDH2 [73,74,75]. Indeed, 10 μM of disulfiram induces apoptosis of TNBC MDA-MB-231 ALDH1A3 overexpressing cells [73]. Similarly, a complex of disulfiram with copper reduced ALDH2 expression in lung cancer A549 cells and enhanced chemosensitivity to taxol [75].

The intricate involvement of ALDHs in cellular homeostasis and metabolism highlights their significance in normal and physiological processes. However, the role of specific ALDH isoforms, such as ALDH1A1, ALDH1A3, ALDH2, and ALDH3A1, as CSC markers has shifted attention towards developing targeted inhibitors to prevent cancer progression. While there is an abundance of evidence highlighting the use of both pan and isoform-specific ALDH inhibitors as anti-cancer agents, none have progressed into clinics, namely, due to working concentrations that surpass those required for clinical translation. The versatility of ALDH inhibitors in targeting multiple cancer types emphasizes their potential as a valuable addition to the armamentarium of cancer therapy, likely as adjuvant agents. The ongoing preclinical research and the development of highly specific and potent ALDH inhibitors will be pivotal in assessing their translational potential.

## 3. EpCAM

Epithelial cell adhesion molecule (EpCAM) is a type I transmembrane protein, and in the early research, it was classified as a cell adhesion molecule (CAM) due to its role in mediating cell–cell adhesion [76]. While most CAMs display broad tissue distribution, EpCAM is restricted to epithelial tissue in healthy individuals and is generally located on the basolateral or basal cell membrane [77,78]. The role of EpCAM in cancer was first elucidated in 1979, when it was described as a tumor antigen in colorectal carcinomas [79]. Indeed, EpCAM is highly expressed in lung, colon, intestine, breast, and prostate carcinoma [80]. Beyond its proposed role as a CAM, EpCAM has also been implicated as a CSC marker and mediator of cell proliferation, migration, and gene expression in cancer [80,81,82,83]. In breast cancer, EpCAM-positive cells display CSC characteristics, with potential for self-renewal and differentiation, as well an enhanced tumorigenicity and migration [82]. Similarly, isolated EpCAM^+^ HCC cells initiated highly invasive HCC in mice and exhibited similar cancer-stem-cell-like traits [84]. In AML, EpCAM^+^ K562 and HL60 cells are resistant to chemotherapy and display enhanced tumorigenicity, compared to EpCAM^low^ cells [85]. Furthermore, EpCAM^+^ ovarian cancer cells exhibited CSC characteristics of tumorigenicity and migration and were resistant to platinum-based chemotherapy [86,87]. However, increased EpCAM expression has also been linked to improved patient survival in some cancers, including renal clear cell carcinoma and thyroid carcinoma, indicating its role as a cancer promoter or suppressor may be cancer-type specific [88,89]. Regardless, the abundance of evidence implicating EpCAM as a CSC marker and its role in promoting aggressive cancer implicate it as an attractive target for anti-cancer agents. Several agents have been developed, including small molecule inhibitors, immunotherapy agents, monoclonal antibodies (mAbs), and aptamers. Table 1 provides a summary of the inhibitors.

### 3.1. Antibodies

As EpCAM is primarily expressed on the surface of cancer cells, there has been great interest in the development of EpCAM mAbs. EpAb2-6, a highly specific EpCAM antibody, induced apoptosis via inhibition of EpCAM signaling in HCT116, SAS, AsPC-1, and SW620 cells, an effect that was diminished in HCT116/shEpCAM and normal cells [90]. Co-treatment of EpAb2-6 (20 mg/kg) alone and in combination with IFL (irinotecan, leucovorin, and fluorouracil (5FU)) in mice bearing colon cancer xenografts significantly decreased tumor volume and improved survival times [90]. In a similar study, EpAb2-6 (20 mg/kg) in combination with atezolizumab, an anti-programmed cell death ligand 1 (PD-L1) antibody, nearly eradicated tumors in peripheral blood mononuclear cell (PBMC)-H441-xenografted mice and increased the CD8^+^ T cell infiltration compared to mice treated with atezolizumab or EpAb2-6 alone [91]. Adecatumumab (MT201) is a human anti-EpCAM mAb that induces cell death via antibody-dependent cellular toxicity and displays intermediate affinity for EpCAM, which is generally thought to be more well-tolerated than high-affinity antibodies [92]. EpCAM^+^ ovarian cancer cell lines established from patients with chemotherapy-resistant disease (OSPC-ARK-1, OSPC-ARK-2, OSPC-ARK-4, CC-ARK1, and CC-ARK-2) treated with 5 μg/mL of adecatumumab displayed increased cytotoxicity compared to control [92]. Adecatumumab has also been tested in several Phase I and II clinical trials and is generally well-tolerated [93,94,95]. In breast cancer patients, adecatumumab reduced the incidence of metastasis in patients with high EpCAM expression but did not result in partial or complete response in any of the patients [93]. Similarly, adecatumumab delayed disease progression in prostate cancer patients with EpCAM^+^ tumors and rising prostate-specific antigen (PSA) levels after prostatectomy [94]. A novel anti-EpCAM antibody, AM-928, is currently recruiting for the first-in-human Phase-I study (NCT05687682) [96]. Recently, there has been increasing interest in the use of bispecific T-cell-engaging (BiTE) antibodies to facilitate T-cell recruitment and activation [97]. Solitomab (MT110) is a BiTE antibody construct that binds to EpCAM and CD3 [98]. Indeed, pancreatic cancer cells AL6 and 185, co-cultured with (PBMCs) and 100 ng/mL of solitomab, significantly reduced spheroid formation compared to PBMCs alone [98]. Furthermore, in vivo treatment of solitomab in mice with PBMC co-cultured A6L-derived tumors resulted in complete abrogation in tumor forming ability, indicating that the tumorigenic CSC population had been effectively eradicated [98]. Ovarian cancer cell lines with high EpCAM expression have been shown to be resistant to T-cell-mediated killing; however, following incubation with solitomab (1 μg/mL), they became highly sensitive to T-cell-mediated killing [99]. Despite promising results in vitro and in murine models, solitomab has displayed relatively modest results in humans. Indeed, in a phase I trial for solitomab on refractory solid tumors, the best outcome among patients was confirmed stable disease (NCT00635596) [100]. Catumaxomab is a similar BiTE antibody construct targeting EpCAM and CD3 that was initially approved in the European Union in 2009 for intraperitoneal treatment of malignant ascites but was withdrawn for commercial reasons in 2017 [101]. Interestingly, a recent paper on mRNA-lipid nanoparticle (LNP) delivery of EpCAM-CD3 BiTEs shows promise in vivo [102]. Intratumoral injection of EpCAM-CD3 hFc mRNA-LNPs (1 μg/mouse) in combination with intravenous injection of T cells significantly decreased OVCAR-5 tumor growth in mice [102].

### 3.2. CAR-T Therapies

Chimeric antigen receptor redirected T cells (CAR-T) are another promising avenue of targeting EpCAM in cancer. In EpCAM^+^ ovarian cancer cell line SW626, anti-EpCAM DNA CAR-T cells displayed high lysis activity, being able to kill 91.5% of SW626 cells [103]. Furthermore, just one intraperitoneal injection of anti-EpCAM DNA CAR-T cells in mice bearing SKOV3-Luc ovarian tumors resulted in decreased tumor burden and increased survival [103]. Adoptive transfer of peripheral blood lymphocytes (PBL) transduced with an EpCAM-specific CAR prevented metastasis of prostate cancer PC3 cells and significantly reduced tumor growth of PC3M cells in nonobese diabetic/severe combined immunodeficiency (NOD/SCID) mice [104]. A recent publication examined the ability of anti-EpCAM CAR-T cells to prevent lung cancer brain metastasis [105]. While intraparenchymal injection of anti-EpCAM CAR-T cells reduced LL/2 tumor growth in the brain in vivo, intravenous administration of the CAR-T cells failed to accumulate within the tumor and, in both cases, failed to persist within the tumor, suggesting that the success of CAR-T cells may be cancer- and location-specific [105]. Anti-EpCAM CAR-T cells have also been shown to produce robust anti-tumor responses in gastric cancer SNU-638 and MKN-45 and pancreatic Capan2-Fluc^+^ models in mice, with complete responses maintained in nearly all the mice [106]. Despite the relative success of CAR-T cells as anti-cancer agents, cancer relapse may still occur and often manifests in antigen escape [107]. Thus, a novel mechanism to circumvent antigen escape is through bispecific CAR-T cells, which target two antigens. Indeed, bispecific EpCAM and intercellular adhesion molecule-1 (ICAM-1) CAR-T cells completely cleared subcutaneous SNU-638 tumors in mice and prevented relapse more durably than anti-EpCAM CAR-T cells alone [106]. Several clinical trials have begun due to the success of anti-EpCAM CAR-T cells in vitro and in mouse models. IMC001, an anti-EpCAM CAR-T therapy, as monotherapy or in combination with immune checkpoint inhibitors, is currently in Phase I and II trials for EpCAM^+^ tumors of the digestive system (NCT05028933, NCT04196465). The preliminary results released in early 2023 for the phase I trial indicated that of the six gastric cancer patients, two achieved partial response, while three were evaluated as having stable disease [108].

### 3.3. Immunotoxins

In a similar vein, the use of immunotoxins to target EpCAM has also been of interest. VB6-845 is an anti-EpCAM fragment antigen-binding (Fab) region linked to DeBouganin, a plant-derived toxin that inhibits protein translation through inactivating RNA N-glycosidase activity [109]. In vitro, VB6-845 displayed high cytotoxicity against EpCAM^+^ cancer cell lines NIH:OVCAR-3, Caov-3, MCF-7, NCI-H69, HT29, and CAL 27 with nanomolar IC50 values (0.4–1.8 nM). In vivo experiments with NIH:OVCAR-3 tumor bearing mice revealed VB6-845 (10 and 20 mg/kg) has potent anti-tumor activity, significantly decreasing tumor volume and resulting in 100% survival in NOD/SCID mice [110]. In phase I clinical trials (NCT00481936), VB6-485 showed modest efficacy, with decreases in tumors ranging from 4 to 15% in a patient with breast carcinoma and 11 to 29% in a patient with renal cell carcinoma [110,111]. A similar recombinant immunotoxin, VB4-845, is an anti-EpCAM scFv fragment linked to Pseudomonas exotoxin A (ETA) [112]. Intratumoral injections of VB4-845 in combination with Nivolumab reduced tumor growth in mice, likely via the complimentary action of VB4-845-mediated immunogenic cell death (ICD) [113]. In a phase I trial of VB4-845 in patients with recurrent, advanced HNSCC, there were several instances of partial response and at least one patient with confirmed complete response [114]. As of 2008, VB4-845 was reportedly undergoing phase II clinical trials in HNSCC, as well as phase III trials for the treatment of superficial transitional cell carcinoma of the bladder (NCT04859751) [115,116]. A more recently developed anti-EpCAM immunotoxin, scFv2A9-PE, similarly uses the Pseudomonas aeruginosa exotoxin and exhibits an IC50 of 50 pM for cytotoxicity in EpCAM^+^ HHCC hematopoietic cells; however, it has yet to be tested against cancer cells [117].

### 3.4. Pharmacologic Inhibitors

As evidenced, immunomodulatory agents have been of great interest in the targeting of EpCAM to treat cancer. However, relatively few pharmacological inhibitors against EpCAM have been developed. Recently, high-content screening of a small molecule compound library yielded several robust inhibitors of EpCAM signaling [118]. Indeed, several of the compounds reduced EpCAM-mediated cyclin D1 (CCND1) expression in colorectal carcinoma cell line HCT-8; however, they have yet to be tested against tumor growth [118]. Other mechanisms of pharmacological inhibition of EpCAM have targeted EpCAM-related pathways instead. SyntOFF, an inhibitor of the syntenin PDZ2 domain, significantly decreased the loading of EpCAM into exosomes and reduced mammosphere formation in MCF-7 cells at 50–100 μM [119].

### 3.5. Antibody-Drug Conjugates

Interestingly, other groups have used anti-EpCAM antibody–drug conjugates (ADCs) to target cancer cells. The novel therapeutic ADC α-amanitin-glutarate-chiHEA125 (chiHEA125-Ama) is an anti-EpCAM mAb conjugated to the mushroom toxin α-amanitin, which inhibits DNA transcription [120]. Administration of a single intraperitoneal injection of chiHEA125-Ama (50 μg/kg) dramatically decreased pancreatic BxPc-3 tumor volume in mice, resulting in complete tumor regression in 50% of the mice [120]. While tumor relapse was observed in 80% of the mice receiving 50 μg/kg of chiHEA125-Ama, 100 μg/kg of chiHEA125-Ama prevented tumor relapse in close to 40% of the mice tested, suggesting the eradication of tumor initiating cells [120].

### 3.6. Other EpCAM-Targeting Strategies

Recently, there has been increasing interest in the use of aptamers and RNA interference (RNAi) to target cancer. Aptamers are single stranded RNA or DNA oligonucleotides that can bind to targets including proteins, nucleic acids, and organic molecules [121]. Anti-EpCAM aptamers can be used to deliver therapeutics to EpCAM^+^ cells. Interestingly, a comparison between anti-EpCAM antibodies versus anti-EpCAM aptamers revealed that aptamers display better tumor penetration and retention in HT29 tumors in mice [122] Indeed, in a 2016 study, an anti-EpCAM aptamer was used to deliver small interfering RNA (siRNA), which reduced EpCAM mRNA expression and protein levels, as well as reducing viability in breast MCF-7 and retinoblastoma WERI-Rb1 cells [123]. Alternative mechanisms of siRNA delivery include the conjugation of anti-EpCAM mAbs to polyetheleneimine (PEI)-capped gold nanoparticles (AuNP) loaded with EpCAM siRNA [124]. In this study, internalization of the EpCAM antibody-conjugated AuNP-PEI and delivery of the EpCAM siRNA resulted in a significant downregulation of EpCAM gene expression and viability in retinoblastoma Y79 cells [124].

Thus, it is evident that the role of EpCAM in cancer biology extends beyond its initial characterization as a CAM to encompass crucial functions as a CSC marker. The development of diverse agents ranging from mAbs and immunotoxins to CAR-T cells and pharmacological inhibitors reflects the concerted efforts to exploit EpCAM’s significance as a contributor to stemness across multiple cancers. Despite these advancements, the complexity of EpCAM necessitates continued exploration and refinement of EpCAM-targeted therapies for translation into the clinical setting and improving outcomes for patients with EpCAM-expressing malignancies.

## 4. CD44

CD44 is a transmembrane glycoprotein that displays broad tissue distribution and expression. CD44 can bind to several ligands to influence numerous cellular processes, including hyaluronic acid (HA), fibronectin, osteopontin (OPN), collagen, and matrix metalloproteinase-9 (MMP-9) [125,126]. CD44-HA binding induces conformational changes in the CD44 structure, allowing binding of cytoskeletal elements and adaptor proteins, which can lead to activation of signaling pathways involved in proliferation, migration, invasion, and adhesion [125,127]. It is well-appreciated that CD44 is an important prognostic marker for CSCs and is one of the most widely recognizable CSC markers across cancer types. Indeed, CD44 is a marker for CSCs in lung, breast, gastric, liver, CRC and AML cancers [128,129]. CD44 expression has been linked to the epithelial–mesenchymal transition (EMT), which is an important feature of stemness in CSCs [130]. Indeed, CD44^+^ pancreatic ductal adenocarcinoma (PDAC) cells with an EMT phenotype display increased tumorigenicity and resistance to gemcitabine compared to CD44^−^ cells [131]. In 2003, Al-Hajj et al. identified a CD44^+^CD24^−/low^Lineage^−^ of breast cancer cells isolated from breast cancer patients [1]. This CD44^+^ subpopulation displayed incredible tumorigenicity, being able to form tumors in mice with as few as 100 cells [1]. Furthermore, serial passaging of the isolated CD44^+^ population gave rise to not only the highly tumorigenic CD44^+^ cells but also a mixed population of cells that form the bulk of tumors, highlighting the role of CD44 in maintaining features of stemness in CSCs [1]. In gastric cancer cells, short hairpin RNA (shRNA) knockdown of CD44 reduced spheroid formation in vitro, as well as tumorigenicity in vivo [132]. Furthermore, CD44^+^ gastric cancer cells also displayed enhanced resistance to chemotherapy and radiation-induced cell death [132]. Thus, developing therapeutics to target CD44 is of great interest in the pursuit to eliminate CSC populations and improve survival in patients (summarized in Table 1).

### 4.1. Antibodies

Several mAbs against CD44 have been developed as cancer therapeutics. Bivatuzumab (BIWA-4) is an anti-CD44 mAb conjugated to microtubule inhibitor mertansine that was tested in HNSCC patients (NCT02254018) [133]. While three patients experienced partial response, significant skin toxicities with a fatal outcome occurred in one trial, and clinical trials were halted [133]. Pre-treatment of another mAb, H4C4, on pancreatic PANC-1 cells significantly decreased tumor formation and reduced clonogenicity in vitro [134]. Furthermore, in a post-radiation tumor recurrence model, only 25% of mice exhibited recurrent tumor growth when treated with H4C4, compared to 90% of mice treated with radiation alone, indicating that the CD44^+^ tumor initiating population had been eradicated by H4C4 treatment [134]. Interestingly, H4C4 also downregulated the expression of several CSC genes, including *Sox2* and *Nanog* [134]. In MCF-7 cells with acquired invasive potential, CD44 is significantly upregulated [135]. However, treatment with IM7, a CD44 mAb, reduced the invasive and migratory capacity of aggressive MCF-7 cells and MDA-MB-231 cells [135]. In a chronic lymphocytic leukemia model, human leukemia stem cells (LSC) from imatinib-responsive or -resistant patients were transplanted into immunodeficient mice [136]. Treatment with RG7356 (RO5429083), a humanized CD44 mAb, reduced the number of LSCs in bone marrow in both the imatinib-responsive and -resistant mice and sensitized the LSCs to dasatinib, a kinase inhibitor [136]. Interestingly, RG7356 was also shown to induce internalization of cell-surface CD44 in chronic lymphocytic leukemia (CLL) cells, and treatment with 0.01 mg/kg of RG7356 in CLL xenografted mice resulted in complete clearance of the leukemia cells [137]. Furthermore, RG7356 displayed remarkable tumor inhibition in HNSCC CAL 27 xenografts in mice [138]. Interestingly, RG7356 also expanded the NK cell population in PMBCs derived from healthy donors [138]. RG7356 has also been tested in phase I clinical trials for patients with solid tumors (NCT01358903) and AML (NCT01641250). While RG7356 was generally well-tolerated in both studies, the majority of patients experienced cancer progression. In the solid tumor clinical trial, the best response recorded was stable disease, which accounted for 21% of patients enrolled [139]. Conversely, in patients with AML, at least one complete response and partial response were recorded [140].

### 4.2. NIR-PIT

In recent years, anti-cancer therapy has expanded to include near-infrared photoimmunotherapy (NIR-PIT). NIR-PIT induces targeted, selective death in cancer cells through conjugation of a mAb to a photo-absorber, silica-phthalocyanine (IRDye700DX: IR700) dye [141]. Injection of a CD44 mAb-IR700 dye conjugate (anti-CD44-IR700) in mice bearing oral SCC CD44^+^ MOC1 and MOC2 tumors significantly inhibited tumor growth and prolonged survival [142]. Interestingly, a similar study combined anti-CD44-IR700 with a systemic anti-PD-1 mAb, resulting in nearly complete eradication of MC38-luc tumors in 70% of the mice [143]. Moreover, this combination treatment also resulted in control of several distant MC38 tumors, indicating that local treatment of one tumor is sufficient to induce systemic anti-cancer effects [143]. Other studies have combined CD44- and CD25-targeted NIR-PIT to simultaneously deplete FOXP3^+^CD25^+^CD4^+^ Treg and CD44^+^ CSCs in several syngeneic tumor models [144]. Indeed, in MC38-luc-tumor-bearing mice, combined CD44- and CD25-targeted NIR-PIT significantly reduced tumor size and regrowth following treatment [144]. Significantly, in a LL/2 tumor model, just one round of CD44- and CD25-targeted NIR-PIT achieved complete remission in 33% of the mice tested [144]. While clinical trials have begun using cetuximab-IR7000 conjugates (NCT02422979), none have commenced using anti-CD44-IR700 conjugates.

### 4.3. Other CD44-Targeting Strategies

Other interesting mechanisms of targeting CD44 in cancer include recombinant human proteoglycan 4 (rhPRG4). rhPRG4 has shown success in preserving joint health in vivo in osteoarthritis models, and a recent study demonstrated that it can also suppress transforming growth factor beta (TGFβ)-induced migration and invasion in breast cancer cells by reducing CD44 protein levels [145,146]. It is well-appreciated that TGFβ can increase CD44 expression in breast cancer cells through EMT [147]. Indeed, the TGFβ-induced invasive capacity of MDA-MB-231 cells is at least in part mediated by the HA-CD44 signaling axis [146]. Interestingly, rhPRG4 reduced the TGFβ-induced increase in the CD44 protein level, thus, reducing the invasiveness of breast cancer HCC38 cells [146]. Aptamers and antisense oligonucleotides (ASOs) are another avenue of CD44-targeting that has garnered interest over the past decade. DNA aptamers such as AS1411, which targets nucleolin, have been used in clinical trials with some success [148]. A few DNA aptamers against CD44 have been developed including Apt#7, which inhibited migration of breast HCC38 cells in vitro by preventing interactions between CD44 and erythropoietin-producing hepatocellular receptor tyrosine kinase class A2 (EphA2) [149]. Other approaches have utilized bi-specific aptamers that target CD44 and EpCAM [150]. Intraperitoneal injections (2 nmol/mouse/every two days) of the bi-specific CD44-EpCAM aptamer in mice bearing ovarian OVCAR8 tumors resulted in increased cancer cell apoptosis and a significant reduction in tumor growth [150]. ASOs are single stranded deoxyribonucleotides that bind to mRNA targets and can be used to modify protein expression [151]. The novel CD44 ASO, ASO 4401, sensitized hepatocellular carcinoma SNU-449 cells to doxorubicin and decreased their clonogenicity and invasive capacity [152]. Several phase I and II clinical trials have commenced using ASOs as cancer therapeutics; however, the use of CD44-specific ASOs has not yet been investigated [153].

### 4.4. Pharmacologic Inhibitors

Several pharmacological inhibitors for CD44 have been developed as anti-cancer agents. The small molecule 1,2,3,4 tetrahydroisoquinoline (THIQ) is a novel inhibitor with affinity for the HA-binding pocket on CD44 [154]. In HNSCC cell lines, treatment with 1 mM of THIQ reduced CD44 protein levels and CD44^+^ cell population [155]. Interestingly, THIQ treatment also reduced clonogenicity and migratory potential in cisplatin-resistant SCC-131 and CAL-27 cells post cisplatin treatment [155]. Recent modifications of THIQ generated a THIQ-derivative, JE22, which was conjugated to nanoparticles (JE22-NP) [156]. In CD44^+^ MDA-MB-231 breast cancer cells, JE22-NP displayed a cell viability half maximal efficacy concentration of 49 nM, significantly reducing cell viability [156]. Interestingly, JE22-NP did not exhibit cytotoxic effects on CD44^−^ breast cancer cells or non-cancerous cells. Furthermore, JE22-NP decreased the migratory capacity of MDA-MB-231 cells compared to control [156]. However, whether JE22-NP demonstrates anti-tumor effects in vivo has yet to be investigated. Recently, verbascoside, the active component found in plants from Lamiales order, was identified as a novel CD44 inhibitor which prevents CD44 dimerization and the subsequent release of the CD44 intracellular domain (CD44ICD) [157]. In vitro, 15 μM of verbascoside attenuated the colony formation of glioblastoma U251MG cells and decreased expression of CSC genes *Sox2* and *Nanog*, an effect that was not observed in CD44^−^ PD-GBMC cells [157]. In glioblastoma U251MG models in vivo, verbascoside significantly reduced tumor growth and prolonged survival in mice [157].

### 4.5. Peptides

Other CD44 inhibitors include A6 (SPL-108), which is a CD44-specific peptide derived from urokinase plasminogen activator (uPA) [158]. While the exact mechanism of A6 has yet to be defined, it is generally thought to bind to CD44 and modulate CD44-mediated signaling [158,159]. The efficacy of A6 as an anti-cancer agent has been tested in several cancers, including prostate, multiple myeloma, breast, and ovarian [159,160,161,162]. Interestingly, A6 appears to be most effective at reducing the migratory capacity of aggressive cancer cell lines. Indeed, A6 blocked migration of the OVCAR3, OVCAR8, IGROV-1, and MDA-MB-468 cell lines, with IC50 values ranging from 15 nmol/L to 65 nmol/L [159]. Interestingly, A6 appeared to potentiate cell binding to HA, a CD44-mediated process, thus halting cell migration [159]. In a metastatic melanoma B16-F10 model, 100 mg/kg of A6 reduced the number of metastatic lesions in the lungs of mice by 50% compared to control [159]. Similarly, in PC-3 LN4 prostate tumor bearing mice, 25% of mice receiving 25 mg/kg of A6 developed metastatic lesions in the lymph nodes, compared to 71% in the control group [160]. Several phase I and II clinical trials for A6 have begun. In a phase II trial for ovarian, fallopian, or peritoneal cancer, none of the thirty-one enrolled patients exhibited response (NCT00939809) [163]. However, in a separate ovarian cancer trial, A6 treatment was associated with delayed clinical progression following first line chemotherapy treatment [164]. The efficacy of A6 has also been tested in phase II trials for chronic lymphocytic leukemia (CLL); however, this trial was terminated due to slow enrollment (NCT02046928). Recently, A6 has been used as a targeting agent to deliver PEGylated liposomal doxorubicin, which decreased C-26 colon cancer tumor volume in mice [165].

The well-established connection between CD44 expression and CSC characteristics, such as tumorigenicity, metastasis, and resistance to therapy, underscores its significance as a prognostic marker and a potential therapeutic target. Encouraging preclinical results, such as the remarkable tumor inhibition by CD44-targeted NIR-PIT and the efficacy of specific inhibitors in reducing metastatic spread, highlight the potential of these approaches. Overall, the multifaceted roles of CD44 in cancer progression and an important regulator of stemness provide a compelling rationale for the continued research and translating these insights into effective therapeutic strategies for cancer management.

## 5. CD55

CD55 (also known as decay accelerating factor—DAF) is a plasma membrane-bound protein expressed on multiple cell types, including leukocytes, epithelial and endothelial cells, as well as red blood cells. Canonically, CD55 acts as an inhibitor of complement-mediated lysis by degrading the C3 and C5 convertases, which are required for formation of the membrane attack complex [166,167]. In addition, CD55 has a non-canonical function as a CSC-associated factor and marker, promoting cancer progression and chemoresistance in cervical, ovarian, neuroblastoma, breast, and endometrial cancer [166]. Indeed, high CD55 expression in a neuroblastoma is associated with increased sphere-forming efficiency and self-renewal capacity, as well as poor survival outcomes in patients [168]. Similarly, in breast cancer, high CD55 expression in patient tumors is associated with lower relapse-free survival [169]. Furthermore, xenotransplantation of CD55^+^ MCF-7 breast cancer cells into NOD/SCID mice resulted in significantly increased tumor volume compared to tumors derived from a CD55^low^ population [169]. CD55 is also associated with chemoresistance in endometrioid cancer, as cisplatin resistant CP70 cells displayed high CD55 expression [170]. Interestingly, this CD55^+^ population also exhibited an enhanced ability for self-renewal, which is a hallmark of stemness [170]. Similarly in cervical cancer, CD55+ C33A cells display significant sphere-forming and migratory abilities, as well as enhanced tumorigenesis and radioresistance [171]. Thus, the exploration of therapeutic strategies targeting CD55 in cancer opens diverse avenues for potential interventions against this critical factor in tumorigenesis and aggressive disease (summarized in Table 1).

### 5.1. Antibodies

Following the elucidation of CD55 as an important CSC marker, interest in the development of antibodies against CD55 has increased. In 2019, a novel CD55 chimeric mAb displayed efficacy against colorectal cancer cells [172]. This antibody, referred to as anti-CD55, decreased the viability of HT-29 and LoVo cells, likely via activation of the complement system [172]. Furthermore, this anti-CD55 antibody attenuated LoVo tumor growth in mice and decreased in vitro metastatic invasion of DLD-1 cells [172]. In addition, this novel mAb also synergistically enhanced the anti-cancer activity of 5FU in CRC [172]. Another study has reported the use of neutralizing mini-antibodies to CD55 and CD59, MB55 and MB59, respectively, in combination with rituximab, an anti-CD20 antibody in B-cell lymphoma [173]. Interestingly, MB55-MB59 enhanced complement-dependent cytotoxicity and killing of B-cell lymphoma cell lines and significantly improved survival in LCL2 tumor models in mice when used in conjunction with rituximab [173]. In neuroblastoma models, the use of a CD55 neutralizing antibody (CD55 NAb) successfully reduced SKN-BE(2) xenograft tumor growth in mice and improved survival [168]. Interestingly, CD55 NAb also reduced expression of CSC markers NANOG, OCT4, and CD133 in the xenografted SKN-BE(2) tumors, indicating that CD55 targeting may also suppress stemness through multiple mechanisms [168]. In recent years, the use of radioimmunotherapeutics, which involve the radiolabeling of mAbs, has emerged as a new mechanism to target cancer. A novel lutetium-177-labeled chimeric mAb developed against CD55 (177Lu-anti-CD55) was shown to reduce metastatic expansion of H460 cells in a pleural metastatic lung cancer model and improve survival in mice [174]. Interestingly, 177Lu-anti-CD55 displays synergistic anti-cancer effects when used in combination with cisplatin, significantly reducing H460 cell viability in vitro and increasing the survival of advanced pleural metastatic mice in vivo. However, on its own,177Lu-anti-CD55 appeared to be more effective than cisplatin and combinatorial treatment in early pleural metastatic models [174]. The use of bispecific antibodies in the targeting of CD55 has also recently generated interest in the treatment of cancer. With the recent successes of PD-1/PD-L1 targeting, it is unsurprising that some groups have developed bispecific antibodies targeting CD55 and PD-L1. GB262 is a novel PD-L1/CD55 bispecific antibody that has been shown to suppress PANC-1 tumor growth in mice by increasing T cell activation, as well as antibody-dependent cell-mediated cytotoxicity (ADCC) and complement dependent cytotoxicity (CDC) [175]. Interestingly, GB262 binding and internalization also resulted in intracellular PD-L1 degradation [175]. Despite the vast evidence in the literature supporting the use of anti-CD55 antibodies as anti-cancer agents, very few clinical trials have been completed. However, several clinical trials have commenced for the human anti-idiotypic mAb (105AD7), which mimics CD55 and can be used as a vaccine to induce immune responses [176,177]. In a phase I trial in colorectal cancer patients, 105AD7 increased survival to 12 months post diagnosis of advanced disease, compared to 4 months in unimmunized patients [178]. Interestingly, despite producing anti-CD55 immune responses as an adjuvant agent, 105AD7 did not prolong survival in a double-blind colorectal cancer phase II trial [176,179]. Similar trials with 105AD7 in osteosarcoma exhibited moderate anti-cancer ability, prolonging survival in patients who mounted immune responses [177]. PAT-SC1 is an anti-CD55 IgM antibody that has undergone clinical trials. Indeed, in a gastric cancer clinical trial, a single dose of PAT-SC1 prior to resection surgery resulted in prolonged survival outcomes in a ten-year follow-up [180].

### 5.2. Peptides

One mechanism to investigate potential CD55-specific peptide ligands is phage display. Using this technology, Li and colleagues identified a CD55-specific ligand peptide known as CD55sp [181]. Treatment of CD55^high^ cervical cancer cell lines, SiHa and HeLa, with CD55sp revealed inhibitory effects on cell proliferation, with IC50 values of 208.4 μg/mL and 230.3 μg/mL, respectively. Furthermore, CD55sp promoted apoptosis in SiHa and HeLa cells, evidenced by increased TUNEL staining Annexin V-PE/7-AAD flow cytometry. Recently, the targeting properties of CD55sp have been used in C-phycocyanin/carboxymethyl chitosan/CD55sp (C-PC/CMC-CD55) nanoparticles to induce apoptosis and inhibit proliferation (IC50 = 40 μg/mL) in HeLa cells [182]. In vivo, C-PC/CMC-CD55 nanoparticles effectively accumulated in HeLa tumors and significantly attenuated tumor growth compared to control. Interestingly, ELISA assays revealed that administration of C-PC/CMC-CD55 in mice increased levels of interleukin (IL)-6 and tumor necrosis factor alpha (TNFα), while reducing levels of TGFβ, suggesting at least some level of immune modulation as a possible explanation for the reduction in tumor volume observed in vivo.

### 5.3. Pharmacologic Inhibitors

Compared to the wealth of anti-CD55 antibodies that have been developed as anti-cancer agents, relatively few pharmacological inhibitors have been investigated. Indirect mechanisms of CD55 inhibition include the targeting of prostaglandin D2, which has been shown to induce CD55 expression in a cyclic AMP (cAMP)-dependent manner [183]. The novel indole compound AWT-489 is a prostaglandin receptor (DP) antagonist, thus, reducing the production of cAMP and CD55 [184]. The treatment of colon cancer cell line, LS174T with 10 μM of AWT-489 reduced CD55 levels [184]. However, whether AWT-489 displays significant anti-cancer effects has yet to be investigated.

### 5.4. Other CD55-Targeting Strategies

Additionally, inhibiting CD55 through genetic means such as CRISPR/cas9 and siRNAs has been proposed. CRISPR/cas9-knockout of CD55 in cervical cancer C33A cells suppressed their proliferative and self-renewal capabilities, sensitized cells to radiation, and decreased tumorigenicity in vivo [171]. Lipoplex-mediated delivery of siRNAs against CD55, CD46, and CD59 sensitized breast cancer BT474 and SKBR3, ovarian cancer SKOV3, and lung cancer Calu-3 cells to trastuzumab and pertuzumab [185]. Despite minimal sensitizing effects on its own in BT474, SKOV3, and Calu-3 cell lines, CD55 siRNA alone sensitized SKBR3 cells to the combined trastuzumab and pertuzumab treatment. Interestingly, siRNA-mediated inhibition of CD55 alone increased C3d deposition on all four cell lines, which is an important mediator of opsonization for macrophages. While the effect of CD55 knockdown alone on macrophage-mediated cytotoxicity was not investigated, combined CD55, CD46, and CD59 knockdown in opsonized BT474 cells significantly increased macrophage-mediated cytotoxicity [185].

In conclusion, CD55 has been shown to promote cancer stem cell characteristics in multiple forms of cancer, highlighting its significance in tumorigenesis and aggressive disease. The exploration of therapeutic strategies, particularly the development of anti-CD55 antibodies has shown promising results in preclinical studies and clinical trials. However, despite the encouraging results, challenges such as off-target effects and the pivotal role of CD55 in the complement system necessitate ongoing research to refine and optimize CD55-targeted therapies.

## 6. CXCR4

The chemokine C-X-C receptor 4 (CXCR4) is an essential axis in normal development and immunity. Its cognate ligand is C-X-C ligand 12 (CXCL12), also referred to as stromal cell-derived factor 1 (SDF-1). The CXCL12-CXCR4 axis is evolutionarily conserved, as knockout is embryonic lethal [186]. In the context of the immune system, CXCR4 plays an important role in the migration of lymphocytes by activating proteins involved in cell arrest and adhesion to the endothelium and migration into tissues, such as integrins [186]. In tissue injury, pro-inflammatory damage-associated molecular patterns are released, such as high mobility group box 1 (HMGB1). HMGB1 binds to CXCL12, leading to a conformational change that enhances its affinity for CXCR4, thereby enhancing the migration of immune and mesenchymal cells [187]. CXCR4 is commonly expressed on hematopoietic, endothelial, and stem cells, which will traffic to sites based on concentration gradients of CXCL12 release [188]. Importantly, the CXCR4-CXCL12 axis is responsible for the recruitment of stem cells during wound healing [189]. CXCR4 has an important role in the homing of progenitor cells to the bone marrow; thus, a blockade of CXCR4 is used when mobilization of cells to the periphery is desired (e.g., in some blood cancers like non-Hodgkin’s Lymphoma) [186].

CXCR4 tends to be aberrantly expressed in many cancer types. High CXCL12/CXCR4 may be predictive of poor prognosis in many cancer types as it is associated with higher stage and metastasis and lower survival [190,191]. Furthermore, CXCR4 stimulation by CXCL12 leads to the downstream activation of pathways like mitogen-activated protein kinases (MAPK), phosphatidylinositol-3-kinase (PI3K) pathway activation, and JAK/STAT pathways, ultimately leading to the homing of stem cells, hematopoietic development, angiogenesis, and tumor progression [186]. As CXCR4-CXCL12 is essential for cell migration during embryonic development, including organogenesis and vascularization, it could be a mechanism of inducing a less differentiated phenotype and contribute to metastasis [188]. Thus, it is unsurprising that CSCs tend to overexpress CXCR4, which permits their migration to target areas of high CXCL12 secretion. In gastric cancer, high CXCR4 expression is an indicator of poor prognosis for survival [192]. Indeed, CD44-positivity in gastric CSCs is associated with enhanced metastatic ability, as well as an ability to differentiate and recapitulate a heterogeneous tumor [193]. Similarly in prostate cancer, isolated CXCR4^+^ cells displayed a 3.8-fold increase in tumorigenicity, and tumor growth in vivo compared to CXCR4^−^ cells [194]. Ovarian cancer cell lines with high CD44 expression exhibit increased resistance to cisplatin, and limiting dilution assays revealed enhanced tumorigenicity with as few as 1000 cells [195]. Similarly in breast cancer, CXCR4 was shown to maintain a tamoxifen-resistant CSC population in MCF-7 cells [196]. Thus, targeting CXCR4 is being explored as a viable option as an anti-CSC therapeutic. Drugs targeting CXCR4 are used in clinical practice for various settings, including human immunodeficiency virus and the mobilization of hematopoietic stem cells for transplantation; however, there has been a renewed focus on the targeting of CXCR4 in cancer [186,197,198]. Table 1 provides a summary of the inhibitors.

### 6.1. Antibodies

Several antibodies have been developed to target CXCR4 in cancer and other diseases. Ulocuplumab (BMS-936564/MDX-1338) is a human IgG4 mAb specific for CXCR4 that blocks the binding of CXCL12. In chronic lymphocytic leukemia, only CXCR4^+^ cell lines exhibited increased apoptosis when treated with ulocuplumab [199]. Similarly, in a preclinical study of AML, treatment with ulocuplumab resulted in an induction of apoptosis in a panel of cell lines [198]. Ulocuplumab also exhibited potent tumor growth inhibition in mouse models of lymphoma, multiple myeloma, AML and breast cancer [198,200]. Interestingly, CXCR4 overexpression in multiple myeloma cells leads to the acquisition of a mesenchymal EMT phenotype, a feature that is frequently associated with stemness [201]. Treatment with ulocuplumab inhibited the EMT phenotype in multiple myeloma tumors, decreased tumor volumes, and synergized with another standard chemotherapy, lenalidomide [201]. In a clinical setting, ulocuplumab synergizes with standard of care chemotherapies. A phase I trial of ulocuplumab combination therapy with ibrutinib in Waldenström macroglobulinemia demonstrated an overall response rate of 100%, with 33% of patients achieving a very good partial response (VGPR) [202]. Another clinical trial in multiple myeloma demonstrated that the combination of ulocuplumab with standard chemotherapies resulted in a progression-free survival rate greater than results in the standard therapy alone from previous trials, indicating that anti-CXCR4 treatment may enhance the efficacy and longevity of the standard treatment response in hematological cancers [203]. Ulocuplumab is currently being tested in clinical trials for solid and liquid cancers (NCT01120457, NCT02472977, NCT02305563, NCT03225716, NCT01359657, NCT02666209).

Other anti-CXCR4 antibodies such as PF-06747143 have demonstrated reduced cancer burden and migration and synergized with standard-of-care chemotherapies in preclinical models of hematological malignancies [204,205]. In preclinical models of chronic lymphocytic leukemia, PF-06747143 induced cell death with high specificity to cancer cells, synergized with standard of care chemotherapy drugs, and improved the survival of treated mice [206]. Additionally, PF-06747143 is significantly more effective than plerixafor, a pharmacological CXCR4 inhibitor, at reducing CXCL12-induced migration and actin polymerization [206]. PF-06747143 is in an ongoing trial (NCT02954653) but there are no results yet posted. Similarly, 12G5 is an anti-CXCR4 antibody that has demonstrated anti-tumor and anti-metastatic activity in both endometrial cancer and osteosarcoma xenografted mouse models [207,208]. Nanoantibodies such as ALX-0651 have also been investigated to target CXCR4. In a phase I study in healthy volunteers (NCT01374503), ALX-0651 demonstrated appreciable CXCR4 targeting ability; however, the study and any subsequent investigations into ALX-0651 as an anti-cancer agent were halted due to an inability to outcompete other CXCR4 antibodies [209]. Other anti-CXCR4 antibodies include LY2624587, hz515H7 (F-50067), and MEDI3185, which exhibit anti-tumor activity in hematological malignancies in mice [210,211,212]. LY2624587 (NCT01139788) and hz515H7 have both been tested in Phase I clinical trials. While results for LY2624587 have not yet been posted, hz515H7 in combination with lenalidomide and dexamethasone resulted in a 66.7% response rate in multiple myeloma patients [213].

### 6.2. Pharmacologic Inhibitors

Several CXCR4-antagonist small-molecule compounds have also been investigated, with Plerixafor (AMD3100, Mozobil^®^) being approved by the US Food and Drug Administration (FDA) in 2008 for multiple myeloma and lymphoma. Plerixafor is a first generation CXCR4 inhibitor and is the most studied and widely used CXCR4 inhibitor, with significant evidence establishing its efficacy as an anti-cancer agent in many cancer types [214,215,216]. Preclinical studies in breast, lung, colon, prostate, and pancreatic cancer showed that CXCR4^high^ tumors have enhanced metastatic potential compared to CXCR4^low^, and plerixafor as a monotherapy or combined with chemotherapy successfully reduces tumor growth and metastases through inhibition of the extracellular signal-regulated kinase 1/2 (ERK1/2) and AKT pathways [214,216,217,218,219,220,221]. Zhou et al. found that CXCR4 may be a key player in tamoxifen resistance in breast cancer, as resistance is reversed when CXCR4 is knocked down or upon treatment with plerixafor due to inhibition of the AKT phosphorylation pathway [190]. In vivo, plerixafor and tamoxifen significantly reduced tumor growth compared to tamoxifen alone, with a decrease in tumor CXCR4 expression [190]. Co-culture of PC3-luc prostate cancer cells with CXCL12-producing stromal cells induces resistance to the chemotherapy drug docetaxel and protects the cells from apoptosis; however, when they are treated with docetaxel and plerixafor, they become sensitive. The combination therapy has a synergistic effect in vivo with reduced tumor burden and metastasis [221]. Treatment with plerixafor in a mouse model of metastatic breast cancer revealed that plerixafor reduced stromal cancer-associated fibroblasts—which play an important role in drug resistance and immune exclusion, reduced hypoxia, and fibrosis-related genes like *Tgfβ1*, while increasing the expression of activated immune-related genes like interferon gamma (IFNγ) and granzymes A and B [222]. Plerixafor synergized with anti-PD-1 and anti-CTLA-4 to improve survival and reduce metastases in mouse models of breast cancer [222]. These studies implicate CXCR4 as a key player in metastasis and drug resistance that can be overcome with CXCR4 inhibitors.

Finally, plerixafor has demonstrated specific effects on CSCs. Heckmann et al. hypothesized that the CXCR4 axis supports drug resistance by inducing dormancy associated with CSCs, therefore disrupting CXCR4 signaling with plerixafor overcomes resistance by mobilizing dormant cells from their protective niches [216]. Colon cancer cell lines with high CXCR4 are inherently more stem-like and susceptible to cytostatic drugs compared to their low expression counterparts. Addition of plerixafor increased the sensitivity of these stem-like cells to the cytostatic chemotherapy drugs tested [216]. Mobilization of quiescent stem-like cells into the periphery from their protective niches using CXCR4 antagonists should make them more susceptible to chemotherapy and aid in overcoming drug resistance.

Other CXCR4 antagonists are being generated and tested. Mavorixafor (X4P-001) is an oral allosteric CXCR4 inhibitor currently in clinical trials: NCT02823405, NCT05103917, NCT02667886, NCT02923531, and NCT04274738. In a phase I trial of clear cell renal cell carcinoma, mavorixafor treatment resulted in a significant decrease in latency-associated peptide of TGF-β1 and angiopoietin-1 from baseline [223]. Another phase I trial tested the efficacy of mavorixafor and pembrolizumab on immune cell infiltration to tumors in melanoma [224]. Combination therapy enhanced the tumor inflammatory score and supported the idea that anti-CXCR4 treatment may reduce immunosuppression in tumors. USL311 is another selective CXCR4 antagonist that is currently undergoing a clinical trial in glioblastoma (NCT02765165). A similar CXCR4 antagonist, PRX177561, in combination with anti-VEGF therapy in glioblastoma, significantly extended the time to progression and disease-free survival in several mouse models [225]. Of note, the authors found that CXCR4 is more highly expressed in glioblastoma CSCs than differentiated cancer cells, and the stem cells are more sensitive to PRX177561 compared to differentiated cells [226]. The CXCR4 antagonist MSX-122 has generated interest as the only oral non-peptide CXCR4 inhibitor. MSX-122 has displayed remarkable anti-metastatic efficacy at low doses (4 mg/kg/day) in several metastatic mouse models of breast cancer, head and neck squamous cell carcinoma, and uveal melanoma [227]. A phase I clinical trial for MSX-122 (NCT00591682) commenced in 2007; however, it was suspended for unknown reasons.

### 6.3. Peptides

The use of synthetic peptides has also generated interest in the targeting of CXCR4. Motixafortide (BL-8040, BKT-140, TNI4001) is a synthetic peptide CXCR4 inhibitor being tested in both solid and liquid cancers. Motixafortide binds the same pocket on CXCR4 as its ligand CXCL12/SDF-1, thereby stabilizing the inactive conformations of CXCR4, preventing its activation [228]. Since Motixafortide acts as an inverse agonist, occupying the binding pocket of CXCR4 without stimulating the receptor, it prevents the rehoming of leukemic blasts to the bone marrow. In a phase IIa trial investigating BL-8040 with high-dose cytarabine in AML, the complete response rate was 29–39% based on dosing regimen (NCT01838395) [229]. Bone marrow aspirates from patients following BL-8040 treatment demonstrated an approximately 73% decrease in CXCR4 BL-8040-occupied AML blasts from pre-treatment levels, indicating that the treatment effectively mobilized cancer cells from the bone marrow to the peripheral blood. Motixafortide induces apoptosis in AML by upregulation of miR-15a/miR-16-1, which resulted in the downregulation of antiapoptotic genes *Bcl-2*, *Mcl-1*, and *cyclin D* [229,230]. The mobilization of cells from the bone marrow to the periphery has important implications in the treatment of blood cancers, as the mobilization of quiescent stem-like cells into the blood stream from their protective niches using CXCR4 antagonists should make them more susceptible to chemotherapy.

Motixafortide has demonstrated success in the treatment of solid tumors. Lefort et al. used breast cancer patient-derived xenografts (PDXs) to test the efficacy of motixafortide compared to the first generation CXCR4 inhibitor, plerixafor, to decrease tumor growth and metastasis [218]. Interestingly, motixafortide demonstrated significant tumor inhibition in human-epidermal growth factor receptor 2 overexpressing (HER2+) breast PDX tumors but was not effective at inhibiting the TNBC PDXs. Furthermore, two trials have been conducted investigating motixafortide in pancreatic cancer. Due to the connection between CXCR4 and immunomodulation, the COMBAT trial (NCT02826486) found that the combination of BL-8040 and pembrolizumab therapy resulted in a rapid and significant increase in leukocytes and lymphocytes to the peripheral blood, with a notable decrease in anti-inflammatory regulatory T cells [231]. The disease control rate of combined BL-8040 with pembrolizumab was 34.5% in PDAC, with increased infiltration of CD8^+^ cytotoxic T cells in the tumor. The authors noted that this study did not have a chemotherapy backbone, so while the results are modest, they do suggest that BL-8040 not only enhanced pembrolizumab response in pancreatic cancer but may expand the benefits of standard chemotherapy treatment [231]. In the subsequent COMBAT trial in pancreatic cancer (NCT02826486), motixafortide and pembrolizumab were used in combination with chemotherapy. Combination with a chemotherapy regimen resulted in an overall response rate of 21.1% with a disease-control rate of 63.2% for a population of patients with poor prognosis and aggressive disease [232].

LY2510924 is a peptide antagonist of CXCR4 being tested in ongoing clinical trials: NCT02737072, NCT02652871, NCT01391130, NCT01439568. Thus far, it has demonstrated limited clinical success in tumor reduction [233,234,235]. In AML, LY2510924 caused a dose-dependent increase in mobilization of AML blasts to the peripheral blood from baseline (>20 fold-change) within one cycle of treatment [235]. A similar peptide CXCR4 antagonist, CTCE-9908, significantly reduced the tumor burden in MDA-MB-231 breast cancer tumor-bearing mice and displayed a 20-fold reduction in metastasis compared to control [236]. CTCE-9908 has been investigated in one phase I/II clinical trial for solid tumors, with the best response being stable disease [237]. More recently, the development of IS4, a novel competitive CXCR4 antagonist, reduced CXCL12-induced migration of prostate and melanoma cell lines more effectively than plerixafor [238].

These studies highlight the ongoing investigation of novel CXCR4 antagonists to treat various cancer types. In sum, CXCR4 expression is associated with worse survival and is overexpressed in many cancers. It promotes cancer by enhancing proliferation, migration, and invasion; promotes metastasis by homing mesenchymal cells to protective niches; and is immunosuppressive. There is significant clinical evidence to support the use of CXCR4 inhibitors to target these mechanisms and safely synergize with standard-of-care treatments in both hematological and solid malignancies. Next generation antagonists, such as motixafortide, are exhibiting enhanced efficacy at very low doses and are well-tolerated in clinical practice. These inhibitors also have specific anti-cancer effects on cancer stem-like cells that are key drivers of metastasis, drug resistance, and cancer recurrence.

## 7. CD133

CD133 (also known as prominin-1) is a pentaspan membrane glycoprotein that was first identified on murine neural stem cells and human hematopoietic stem cells but has since been found on epithelial cells in multiple tissue types [239]. CD133 is preferentially expressed on microvilli, as well as plasma membrane protrusions such as filopodia, and has also been identified as a cholesterol-interacting protein [239,240]. Interestingly, some evidence suggests that cholesterol microdomains labeled with CD133 preserve stem cell characteristics by inhibiting differentiation [241]. CD133 is also important for the proper morphogenesis of rod photoreceptor cells and has thus been suggested to play a role in proper retinal development [242]. Due to its well-appreciated role as a stem cell marker, it is unsurprising that CD133 is a well-characterized CSC marker as well [239,240]. CD133 was first used to identify CSCs in brain tumors, including glioblastoma and, later, many other cancers [240,243,244,245,246]. In prostate cancer, CD133^high^ cells exhibit enhanced self-renewal and differentiation capabilities. Indeed, isolated CD133^high^ cells gave rise to a phenotypically heterogeneous population in culture [247]. Similarly in liver cancer, CD133^+^ cells possess greater ability to form tumor spheroids in vitro and exhibited enhanced tumorigenicity in vivo [248]. Only CD133^+^ cells sorted from liver tumor xenografts implanted into secondary mouse recipients generated tumors, and tumor growth was enhanced in the secondary tumors compared to primary tumors [248]. In colorectal cancer, CD133^+^ cells isolated from surgically resected tumor specimens generated long-term spheroid cultures that displayed self-renewal and differentiation capabilities, eventually recapitulating the original tumor phenotype [249]. Furthermore, the CD133^+^ spheroid cells were resistant to irinotecan, a further indication of chemotherapy resistance associated with stemness [249]. CD133 has also been identified as a CSC marker in lung cancer. Patient-derived CD133^+^ lung cancer spheroids displayed enhanced tumorigenicity in vivo, and immunohistochemical analysis revealed that the subsequent CD133-derived tumors reproduced a heterogeneous phenotype resembling that of the original patient tumor [250]. High CD133 expression is associated with enhanced clonogenicity, invasiveness, and resistance to chemotherapy in oral squamous cell carcinoma [251]. Clinically, CD133-positivity is associated with aggressive disease and poor prognosis in colorectal and invasive breast cancer [252,253]. However, the role of CD133 as a CSC marker is not without controversy. In renal cell carcinoma, high CD133 expression is associated with prolonged survival and is a marker for good prognosis [254]. Conversely in pancreatic adenocarcinomas, CD133-positivity does not confer significantly enhanced tumorigenicity compared to other CSC markers [255]. Thus, the role of CD133 as a CSC marker is context- and tumor tissue-type dependent. Regardless, given the role of CD133 as a robust CSC marker across multiple different malignancies, targeting CD133 presents an interesting avenue to disrupt stemness and improve patient outcomes (summarized in Table 1).

### 7.1. Antibodies

Several anti-CD133 mAbs have been developed, with many serving as conjugates to nanoparticles. Indeed, conjugation of an anti-CD133 mAb to PEGylated gold nanoparticles was used to target 5-FU delivery in colorectal cancer HCT116 cells [256]. Similarly, targeted paclitaxel delivery to breast cancer MCF-7 cells with anti-CD133 mAb targeting nanoparticles significantly reduced colony-forming potential and mammosphere formation compared to paclitaxel alone [257]. Two injections of the CD133-targeting paclitaxel nanoparticles resulted in a 70% decrease in MDA-MB-231 tumor volume [257]. Another study shows that improving the targeted delivery of the topoisomerase inhibitor SN-38 by incorporating it into PEG–PCL-based nanoparticles with anti-CD133 mAb increases the cytotoxic effects of SN-38 toward colorectal cancer cells and tumor xenografts [258]. Other groups have combined antibody and cellular immunotherapies, generating bispecific anti-CD133/CD3 antibodies bound to cytokine-induced killer cells (BsAb-CIK) [259]. Intraperitoneal injections of BsAb-CIK in pancreatic SW1990-tumor-bearing mice resulted in a significant decrease in tumor growth. This finding suggests that antibody-based targeting of CD133 may hold promise in developing targeted immunotherapies. Indeed, an Fc-optimized CD133 mAb, known as 293C3-SDIE, induces NK cell activation, degranulation, secretion of IFN-γ, and antibody-dependent cellular cytotoxicity, resulting in potent lysis of colorectal cancer cells Caco-2 and HCT-116, as well as B cell acute lymphoblastic leukemia cells, highlighting its potential in solid and hematologic malignancies [260,261].

### 7.2. CAR-T and NK Cells

Other mechanisms of targeting CD133 include modulation of immune cells such as T and NK cells. Indeed, CAR-based therapies targeting CD133 have been developed in recent years. Anti-CD133-CAR expressing NK92 cells displayed specificity against patient-derived CD133^+^ ovarian cancer cells, resulting in reduced cell viability [262]. Furthermore, the co-incubation of cisplatin and anti-CD133-CAR NK92 cells on ovarian cancer cells resulted in increased cell death compared to cisplatin alone, implicating CD133-directed NK cell killing as a possible adjuvant agent for cisplatin-based regimens. More recently, anti-CD133 CAR-T cells have been tested in vivo in small cell lung cancer models (SCLC) [263]. While adoptive transfer of the anti-CD133 CAR-T cells alone reduced tumor burden and improved survival in SCLC mouse models, combined anti-PD-1 therapy, CD73 inhibition, and CD133 CAR-T cells resulted in complete remission for 25% of the mice [263]. These findings add further evidence, suggesting that combination therapy is most effective at targeting heterogenous tumor populations. At least two clinical trials for anti-CD133 CAR-T cells have been completed. Indeed, the findings of a phase I clinical trial (NCT02541370) indicated that infusion of autologous CAR-T-cell-directed CD133 (CART-133) inhibits disease progression in patients with advanced metastatic lung, liver, and colorectal carcinoma [264]. Another phase II clinical trial (NCT02541370) using anti-CD133 CAR-T cells in hepatocellular carcinoma resulted in a median progression-free survival (PFS) of 6.8 months and overall survival of 12 months; however, only one patient achieved partial response [265].

### 7.3. BiKEs and Vaccines

Bispecific killer engagers (BiKEs) have also been used as anti-CD133 agents in cancer. One such BiKE combines CD16 and CD133 single chain variable fragments (scFv) to engage NK cells (16 × 133 BiKE) [266]. Indeed, 16 × 133 BiKE was shown to activate resting NK cells to induce IFN-γ production and degranulation against CD133^+^ colorectal cancer Caco-2 cells, which are known for being resistant to NK-mediated killing. Trispecific (CD16, IL-15, CD133) and tetraspecific (CD16, IL-15, EpCAM, CD133) NK cell engagers have also demonstrated appreciable anti-cancer efficacy against colorectal Caco-2 cells, as well as a panel of breast, prostate, HNSCC, CRC, and AML cell lines, respectively [267,268]. Interestingly, a dendritic cell vaccine against CD133 was tested in a phase I clinical trial (NCT02049489) for recurrent glioblastoma [269]. In this trial, autologous dendritic cells were loaded with CD133 antigen, and while the results indicated that ICT-121 is safe and induces immune responses, it is unclear whether this treatment provided any anti-tumor benefit.

### 7.4. Immunotoxins

Like many other anti-CSC targeted therapeutics, immunotoxins have also played a role in CD133-targeted therapy. GMI, which is a fungal immunomodulatory protein derived from *Ganoderma microsporum*, reduces CD133 protein expression via autophagic-lysosomal protein degradation [270]. Indeed, oral gavage of GMI demonstrated remarkable inhibition of tumor growth in lung pemetrexed-resistant CD133^+^ A549/A400 tumor-bearing mice and significantly decreased CD133 expression in the tumor, indicating good tumor-penetrating capabilities. Other groups have conjugated anti-CD133 scFv to a truncated form of Pseudomonas exotoxin A (PE38), generating the novel immunotoxin dCD133KDEL [271]. dCD133KDEL has been tested against several forms of cancer, including HNSCC, breast, and ovarian, demonstrating appreciable anti-tumor effects [271,272,273]. Pre-treatment of HNSCC CD133^+^ UMSCC-11B cells with dCD133KDEL significantly reduced tumorigenicity in xenotransplantation mouse models, suggesting eradication of the aggressive CD133^+^ CSC population [271]. Bostad et al. reported success using an anti-CD133 mAb conjugated to saporin, a plant-derived toxin in the ribosome inactivating protein family [274]. Interestingly, this group used photochemical internalization to deliver AC133-saporin to WiDr colon cancer tumor-bearing mice, resulting in a significant decrease in tumor growth compared to control [274].

### 7.5. Pharmacologic Inhibitors

Several pharmacologic inhibitors, including small molecule inhibitors and repurposed drugs, have generated interest as CD133 inhibitors. Celecoxib, which is a non-steroidal anti-inflammatory drug, inhibits the expression of CD133 in HT29 and DLD1 colorectal cancer cells via suppressing the Wnt signaling pathway [275]. Conversely, other studies have suggested that celecoxib-induced downregulation of CD133 is mediated by AKT inhibition in HT29 cells [276]. Trifluridine, a drug that is currently in clinics for the management of refractory colorectal cancer inhibits spheroid formation in CD133^+^ DLD1 colorectal cancer cells, suggesting a potential mechanism for its anti-cancer activity [277]. ACT001, a guaianolide sesquiterpene lactone, inhibits stemness and CD133 expression by inducing Olig2 ubiquitination degradation in A549 and NCI-H820 lung cancer cells [278]. However, the exact anti-cancer mechanism of ACT001 remains controversial, as it was identified as a direct inhibitor of plasminogen activator inhibitor-1 (PAI-1) and exerted anti-cancer effects in glioma cells by inhibiting the PAI-1/PI3K/AKT pathway [279]. Regardless of the exact mechanism, ACT001 was approved as an orphan drug by the FDA in 2017, and phase I and II clinical trials have commenced in glioblastoma, with at least one patient achieving complete remission (ACTRN12616000228482, NCT05053880) [280].

### 7.6. Aptamers and Peptides

Aptamers and peptides have also been studied to target CD133. Several CD133-specific aptamers have been used as agents to deliver chemotherapeutics such as doxorubicin. Indeed, CD133 aptamer-guided delivery of doxorubicin results in a higher doxorubicin concentration in Huh7 hepatocellular carcinoma stem cells, resulting in a significant decrease in tumorsphere formation [281]. Other doxorubicin-loaded CD133 aptamers have been tested in vivo, resulting in a significant reduction in diethylnitrosamine-induced liver cancer tumors in mice [282]. Other groups have used CD133 targeted RNA aptamers to deliver an anti-microRNA (miRNA) against miR-21, a well-known miRNA involved in TNBC progression [283]. CD133 aptamer-mediated delivery of anti-miRNA21 into MDA-MB-231 breast cancer tumors in mice resulted in accumulation in the tumor, as well as a significant reduction in tumor growth [284]. A CD133-specific peptide, LS-7, was identified in 2012, which reduced migration and invasion of murine colon and breast cancer cell lines CT-26 and MA-782, respectively [285]. While LS-7 was able to localize to CD133^+^ CT-26 tumors in vivo, its ability to reduce tumor growth has not yet been investigated.

### 7.7. Other CD133-Targeting Strategies

CRISPR/cas9 and siRNAs have also been investigated as a tool to inhibit CD133 expression. Li et al. demonstrated that CRISPR-Cas9 mediated CD133 knockout in colorectal cancer LoVo cells reduced proliferation, colony formation, and EMT, as well as migration [286]. siRNA-mediated suppression of CD133 not only sensitized HT-29 cells to oxaliplatin but also decreased colony formation, migration, and expression of EMT-related genes [287]. In 2016, a study evaluated the effect of CD133 (AC133) mAb-based NIR-PIT on CD133^+^ U251 glioma tumors [288]. In established subcutaneous human xenograft tumors, AC133 NIR-PIT demonstrated significant tumor growth inhibition in CD133-expressing U251 glioma models, confirmed by reduced signal intensity post-NIR light exposure and diminished tumor volumes. Western blot analyses of tumor cell lysates revealed a substantial reduction in CSC markers, indicating specific killing of undifferentiated CSCs and subsequent tumor growth inhibition, as evidenced by various imaging modalities and tumor weight measurements [288].

CD133 serves as a pivotal marker for CSCs across various malignancies, correlating with aggressive disease and poor prognosis in several cancers such as glioblastoma, colorectal, breast, lung, and liver cancers. Various strategies have been explored to target CD133, including antibody-based therapies, CAR-based immunotherapies, pharmacologic inhibitors, immunotoxins, and RNA-based approaches, each showing promising results in preclinical and clinical studies. Further research and clinical trials are warranted to fully explore the therapeutic potential of CD133-targeted interventions and their impact on improving patient outcomes across diverse malignancies.

## 8. Nanog

Nanog is a DNA binding Homeobox transcription factor essential for maintaining pluripotency and self-renewal. Nanog is mainly expressed in pluripotent and developing germ cells and is essential in maintaining stemness (undifferentiated state of the cells), as well as determining embryonic cell fate [289]. Nanog is epigenetically switched off during development. However, in cancers, the Nanog gene is broadly expressed in many cancer types, especially of the epithelial phenotype [290,291,292,293]. Nanog-mediated oncogenic reprograming is observed in malignant cancers and is associated with clinical manifestations of metastasis and secondary tumors [290,292,293,294]. It is well-appreciated that Nanog expression confers a CSC phenotype across multiple cancer types, including liver, colorectal, prostate, breast, ovarian, lung, and brain [293]. Indeed, a meta-analysis of the prognostic significance of elevated Nanog expression in solid tumors revealed that it is associated with more aggressive disease and worse prognosis [295]. Nanog expression is associated with tumorsphere and colony formation, as well as chemoresistance and tumorigenicity in breast cancer [296]. In colorectal cancer, enforced Nanog expression was shown to promote tumorigenicity and generate tumors in mice [297]. Furthermore, Nanog has been identified as a CSC marker in both prostate and HCC cancers [290,298]. Nanog can also modulate expression of other CSC markers, including ALDH, CD44, and CD133, further cementing its role as an important mediator of stemness [290,299,300,301]. Collectively, this makes Nanog a lucrative target to treat advanced cancers.

### 8.1. PMOs and RNA Interference

To date, no drugs that directly target Nanog have been identified, with most chemical inhibitors having a secondary effect of Nanog reduction. Some functional drugs using synthetic DNA and RNAi machinery that directly target NANOG have been identified. Kundu et al., 2020, used synthetic DNA antisense phosphorodiamidate morpholino oligonucleotides (PMOs) targeting NANOG in MCF-7 cells [302]. PMOs are complimentary to the gene start site and block transcription through steric blocking [303]. However, PMOs do not enter intact cells, making their delivery challenging, needing carrier molecules like cell penetrating peptides, nanoparticles, or dendrimers [304]. In this study, the anti-Nanog PMO was conjugated to a nonpeptidic internal guanidinium transporter (IGT) to facilitate intracellular delivery [302]. The IGT-PMO decreased Nanog protein expression in breast cancer MCF-7 cells. Further analysis revealed that Nanog inhibition also reduced clonogenicity, as well as migration and invasion potential, and sensitized MCF-7 cells to Taxol chemotherapy. However, it is important to note that CXCR4 and MMP9 were also downregulated as the result of Nanog inhibition and, thus, may be responsible for the reduction observed in migration and invasion [302]. Nanog knockdown with siRNA has also been shown to increase the sensitivity to 5-FU and decrease self-renewal as well as migration in of CRC SW-480 cells [305].

### 8.2. Pharmacologic Inhibition

Indirect Nanog inhibition is achieved by drugs targeting upstream regulators or those that mediate proteasomal degradation of Nanog. Histone deacetylase (HDAC) has previously been shown to regulate Nanog expression in embryonic stem cells [306]. Suberoylanilide hydroxamic acid (SAHA), which is an FDA-approved drug for T-cell lymphoma as a pan-HDAC inhibitor, markedly reduced Nanog expression in cisplatin-resistant HNSCC cells, sensitizing them to cisplatin [307,308]. In vitro, SAHA reduced CAL27-CisR and UD-SCC-2-CisR tumorsphere formation, and in vivo, significantly reduced tumor growth and metastasis [308]. In both cases, Nanog expression was significantly downregulated. PiB (diethyl-1,3,6,8-tetrahydro-1,3,6,8-tetraoxobenzo [lmn]3, 8 phenanthroline-2,7-diacetate) has also been investigated as an indirect Nanog inhibitor in cancer [309]. PiB is a well-characterized peptidyl-prolyl-cis-trans isomerase NIMA-interacting 1 (Pin1) inhibitor [310,311,312]. Pin1 has been identified as an upstream Nanog regulator by inhibiting speckle-type POZ protein (SPOP)-mediated Nanog polyubiquitination and degradation [309,310]. Indeed, Pin1 inhibition by PiB degrades Nanog though ubiquitination in prostate cancer, resulting in decreased tumorsphere formation and colony growth [309]. Similarly, resveratrol, a phytochemical that has been investigated as an anti-cancer agent in multiple cancer types, reduced Nanog protein levels in glioblastoma by activating proteasomal degradation [313,314]. Treating glioblastoma CSCs with resveratrol followed by intracranial injection into nude mice significantly reduced tumorigenicity and prolonged survival compared to control mice [313]. Interestingly, Nanog has also been identified as a downstream target of aspirin, a well-known non-steroidal anti-inflammatory drug [315]. Indeed, aspirin decreased Nanog protein levels in colorectal cancer HCT116 and LoVo cells by decreasing Nanog stability in a proteasome-independent manner. Consequently, aspirin-treated HCT116 and LoVo cells displayed attenuated stemness features, including tumorsphere formation [315]. Metformin is an FDA-approved first-line drug for diabetes mellitus that has been shown to downregulate Nanog through upstream reduction of kruppel-like factor (KLF5) levels in TNBC [316]. Indeed, metformin suppressed TNBC CSCs at least partially by inhibiting KLF5, evidenced by reduced tumorigenesis in HCC1806 cells implanted into nude mice. In esophageal cancer, iron chelators, DFX and SP10, appeared to suppress the expression of stemness markers, including Nanog [317]. In vivo, SP10 significantly reduced Nanog expression in TE8 tumors and decreased tumor growth.

Thus, Nanog dysregulation in cancer contributes to oncogenic reprogramming, maintenance of pluripotency and self-renewal, and aggressive tumor behaviors. Despite the lack of direct Nanog-targeting drugs, promising avenues for therapeutic intervention have been explored. From synthetic DNA and RNAi approaches to indirect inhibition through targeting upstream regulators or mediators of proteasomal degradation, diverse strategies have shown efficacy in attenuating Nanog expression and mitigating CSC characteristics across various cancer types.

## 9. Notch

Notch1, 2, 3, and 4 are evolutionarily conserved genes involved in various biological processes, including organ development and repair and hematopoietic stem cell differentiation and development [318]. The Notch genes encode four receptors that bind to notch ligands (delta-like 1,4 and jagged 1,2) present in adjacent cells [319]. Upon activation, the notch receptors are cleaved by ADAM family metalloproteases or gamma-secretase, then the notch intracellular domain (NICD) is internalized to the nucleus, resulting in transcriptional regulation [318]. Canonical Notch signaling is different from other receptor mediated signaling as it does not have signaling intermediates [320,321]. While Notch has been identified as a tumor suppressor in some cancers, it has also been implicated as an important contributor to maintaining stemness in CSCs [318]. Indeed, Notch signaling maintains the CSC population in patient-derived glioma cells, evidenced by increased tumorsphere formation and an ability to self-renew and differentiate, effects which were abrogated upon Notch inhibition [322]. In renal cell carcinoma, Notch signaling was upregulated in CD24^+^/CD133^+^ CSCs, and their stemness was at least in part due to Notch activation and signaling, highlighting the role of Notch in promoting expression of other CSC markers [323]. Similarly, Notch expression was upregulated in CD44^+^/CD24^+^/ESA^+^ pancreatic CSCs [324]. Notch pathway inhibition in this CSC population significantly attenuated tumorsphere formation and tumorigenicity in mice [324]. Notch reporter systems have also identified Notch as a major contributor to CSC populations in breast cancer. Indeed, Notch^+^ cells were able to generate heterogeneous Notch^+^ and Notch^−^ SUM159 cells, an effect that was not observed in Notch^−^ cells [325]. Moreover, Notch^+^ SUM159 and MCF-7 cells exhibited increased tumorigenicity in limiting dilution xenograft assays. Thus, targeting Notch signaling presents a promising therapeutic strategy (Figure 2, Table 1).

### 9.1. Pharmacologic Inhibitors

One of the major mechanisms of chemical Notch inhibition is gamma-secretase inhibitors (GSI). Indeed, GSIs are well-studied for Notch intracellular domain (NICD) internalization resulting in notch inhibition. However, one of the pitfalls of Notch inhibition with GSIs is that this class of drugs exhibit significant gastrointestinal toxicity, which limits their clinical translatability. Regardless, several GSIs have been developed, including crenigacestat (LY3039478), LY900009, osugacestat (AL101, BMS-906024), RO4929097 (RG473), and nirogacestat (PF-03084014) [326]. Crenigacestat is a small molecule first generation GSI exhibiting potent NICD inhibition, with a reported IC50 of approximately 1 nM in vitro [327]. Several preclinical studies have demonstrated that crenigacestat is a robust anti-cancer agent with oral bioavailability against breast, colon, lung, ovarian, glioblastoma, gastric, and intrahepatic cholangiocarcinoma PDX models in mice [327,328]. Due to its preclinical success, several clinical trials have been completed investigating crenigacestat in humans. A phase I trial of crenigacestat monotherapy (NCT01695005) in patients with advanced metastatic cancer resulted in modest anti-cancer effects, with one breast cancer patient achieving partial response and ~31–35% of patients achieving stable disease [329]. Other clinical trials have shown that crenigacestat exhibits limited clinical activity in adenoid cystic adenocarcinoma (NCT01695005) and advanced solid tumors (NCT02836600) [330,331]. Interestingly, B cell maturation antigen (BCMA) directed CAR-T cells in combination with crenigacestat appears to demonstrate appreciable anti-cancer efficacy in patients with multiple myeloma (NCT03502577) [332]. LY900009 is a similar GSI that inhibits NICD cleavage and displays an IC50 of 0.005–20 nM [333]. However, in a phase I clinical (NCT01158404) trial for advanced cancer, no patients experienced appreciable response to LY900009 beyond stable disease [333]. Osugacestat (BMS-906024, AL101) is a benzodiazepine succinimide GSI with pan-Notch inhibition activity [334]. Potent anti-tumor activity in preclinical testing with osugacestat against breast MDA-MB-468, T-acute lymphoblastic leukemia (T-ALL) and adenoid cystic carcinoma mouse models led to the design of several phase I and II clinical trials [334,335]. In a phase I trial in patients with relapsed T-ALL, 32% of patients displayed a 50% reduction in bone marrow blasts, with two patients achieving CR and PR, respectively [336]. More recently, the results of the ACCURACY trial (NCT03691207) for osugacestat in recurrent/metastatic adenoid cystic carcinoma revealed that 12% of patients experienced partial response, while 57% achieved stable disease [337]. Osugacestat also showed synergistic activity with paclitaxel in NSCLC xenografts, and a phase I trial (NCT01653470) for osugacestat in combination with paclitaxel, 5FU^+^ irinotecan, or carboplatin + paclitaxel has been completed for patients with metastatic solid tumors; however, results have yet to be released [338]. R04929097 is another notch GSI with an IC50 of 4 nM [339]. RO4929097 reduced Notch signaling in melanoma and significantly reduced tumor volume as well as tumorigenicity in serial xenotransplantation models [340]. Phase II clinical trials in metastatic colon cancer (NCT01116687), melanoma (NCT01120275), sarcoma (NCT01154452), and pancreatic adenocarcinoma (NCT01232829) showed limited clinical efficacy [341,342,343,344]. Nirogacestat (PF-03084014) is a GSI with an IC50 of 6.2 nM. Preclinical studies revealed that nirogacestat has anti-CSC properties, reducing self-renewal and proliferation of HCC CSCs and exhibiting potent anti-tumor activity [345]. Interestingly, nirogacestat appears to synergize with docetaxel in prostate cancer, significantly reducing tumor growth in mice, as well as the frequency of CSCs in the tumors [346]. Numerous phase I and II clinical trials have been completed for nirogacestat across multiple cancer types. In a phase II trial (NCT01981551), nirogacestat monotherapy for pre-treated patients with dermoid tumors resulted in a 29% confirmed partial response rate [347]. More recently, phase II (NCT04195399) and phase III (NCT03785964) trials for nirogacestat in desmoid tumors in pediatric and adult populations, respectively, have commenced. Preliminary results from the phase III trial indicate an objective response rate of 41% compared to the placebo [348]. Clinical trials for nirogacestat in combination with docetaxel in patients with advanced TNBC have also been completed, with 16% of patients achieving confirmed partial response and 36% achieving stable disease (NCT01876251) [349].

Other chemical GSIs include DAPT, MRK-0752, and MRK-560. DAPT has been investigated in preclinical studies. DAPT increased the efficacy of cisplatin in osteosarcoma by directly targeting CSCs, resulting in significant tumor control in vivo [350]. DAPT also reduced CD44^+^ cells in gastric cancer and subsequent Wnt/β-catenin signaling, effectively inhibiting tumorsphere formation [351]. DAPT was effective in reducing Notch levels and affecting growth hormone production and tumor growth in growth-hormone-producing adenomas [352]. MK-0752 was originally developed as a treatment for Alzheimer’s disease, targeting the amyloid precursor protein, and has an IC50 of 50 nM [326]. In breast cancer MC1 xenograft models, MK-0752 reduced the CD44^+^/CD24^−^ and ALDH^+^ CSC populations, and serial transplantation confirmed a reduction in tumorigenicity [353]. In a phase I clinical trial of MK-0752 in combination with docetaxel in breast cancer patients, 45% of the evaluable patients achieved a partial response; however, as there was only one arm in this study, it is unclear whether the efficacy was due to docetaxel or MK-0752 [353]. MK-0752 has also been investigated in preclinical models of ovarian cancer, significantly inhibiting ovarian xenograft tumor growth following cisplatin treatment [354]. Phase I and II clinical trials with MK-0752 in advanced solid tumors, PDAC (NCT01098344), and CNS malignancies have been completed with varying degrees of efficacy [355,356,357]. While no measurable responses were observed in pediatric patients with CNS malignancies, MK-0752, in combination with gemcitabine in PDAC, did result in tumor control activity, with 13 patients achieving stable disease and one achieving a confirmed partial response [356,357]. MRK-560 is a second generation GSI that reportedly displays reduced GI toxicity [358]. Indeed, treatment of T-ALL-patient-derived xenografts with MRK-560 significantly attenuated the leukemia burden and improved survival outcomes in mice without any associated GI toxicity [358].

Several other small molecule Notch inhibitors have also been developed. Limantrafin (CB103) is a recently developed small molecule inhibitor that targets the Notch transcription activation complex [359]. In TNBC and T-ALL xenograft models, limantrafin markedly reduced tumor burden and prolonged survival [359]. Limantrafin in combination with paclitaxel or fulvestrant also potently inhibited mammosphere formation in breast MCF-7 and estrogen receptor positive (ER^+^) cells [360]. Moreover, limantrafin synergized with paclitaxel in TNBC HCC1187 xenografts, resulting in reduced tumor growth and delayed tumor recurrence after discontinuation of treatment compared to paclitaxel alone [360]. ZLDI-8 is a novel a disintegrin and metalloproteinase 17 (ADAM-17) inhibitor, which is an enzyme involved in Notch activation and cleavage [361]. Interestingly, ZLDI-8 inhibited the EMT phenotype of NSCLC, CRC, and HCC cells, sensitizing them to chemotherapeutic agents, as well as suppressing metastasis and tumor growth in vivo [361,362,363].

### 9.2. Antibodies

Antibody-based notch inhibition either targets notch ligands or receptors. The mAb 602.101 is specific to Notch1 and inhibited Notch signaling in breast cancer cell line MDA-MB-231 [364]. Furthermore, treating MDA-MB-231 cells with 602.101 decreased the CD24^−^/CD44^+^ CSC population and significantly attenuated primary, secondary, and tertiary mammosphere formation. Preclinical models using an antibody that targets the negative regulatory region of Notch 1 resulted in the effective control of tumor growth in Calu-6 and HM7 xenograft models [365]. Tarextumab (OMP-59R5) targets Notch receptors 2/3 and displays anti-cancer activity against several epithelial cancers, including breast, lung, ovarian, and pancreatic [366]. Limiting dilution assays derived from tarextumab-treated pancreatic PN8 xenograft tumors revealed that tarextumab reduces CSC frequency and tumorigenicity. Furthermore, tarextumab synergized with paclitaxel and gemcitabine to markedly inhibit tumor growth in PN8 xenograft models [366]. Phase I clinical trials with tarextumab (NCT01859741) showed an 84% overall response rate in small cell lung cancers in combination with etoposide and platinum; however, this treatment did not improve PFS, and the trial was thus terminated [367]. While tarextumab monotherapy in patients with solid tumors (NCT01277146) resulted in reduced Notch signaling, no evaluable responses beyond stable disease were observed [368]. Similarly, a randomized phase II trial (NCT01647828) of tarextumab in combination with nab-paclitaxel or gemcitabine did not improve PFS or overall survival in patients with metastatic pancreatic cancer [369]. Brontictuzumab (OMP-52M51) is a Notch1 antibody that has been investigated in several clinical trials. In a phase I trial for solid tumors (NCT01778439), clinical benefit was observed in 17% of patients [370]. In hematologic malignancies, brontictuzumab displayed moderate anti-tumor activity (NCT01778439) [371]. Brontictuzumab has also been investigated in phase I trials for metastatic CRC (NCT03031691), adenoid cystic carcinoma (ACC) (NCT02662608), and lymphoid malignancies (NCT01703572). In 2019, a phase I clinical trial (NCT02129205) with a novel anti-Notch3 antibody-drug conjugate with auristatin, PF-06650808, 50% of breast cancer patients achieved stable disease [372]. Interestingly, among ER^+^ patients, 21% achieved a partial response.

Several anti-DLL4 antibodies have also been developed to target Notch. Enoticumab (REGN421) is an anti-DLL4 mAb that displayed modest anti-tumor activity in phase I clinical trials (NCT00871559) for patients with solid tumors, with 50% of patients having SD, and 2 out of 32 achieving PR [373]. Demcizumab (OMP-21M18) is another anti-DLL4 antibody, which had an overall response rate of 21% in a phase Ib trial (NCT01952249) for patients with ovarian, peritoneal, and fallopian tube cancer when combined with paclitaxel [374]. In a similar trial for NSCLC (NCT01189968), 50% of patients had objective tumor responses when treated with demcizumab in combination with pemetrexed and carboplatin [375]. Several patients experienced significant cardiac toxicity; however, this effect could be attenuated with a truncated dosing regimen. Other clinical trials for demcizumab in combination with various anti-cancer drugs have been completed, including folinic acid, 5-FU, and irinotecan (FOLFIRI) in advanced pancreatic cancer (NCT01189942), pembrolizumab in advanced solid tumors (NCT02722954), and paclitaxel in platinum-resistant ovarian cancer (NCT01952249). MEDI0639 is a novel DLL4 antibody that inhibits angiogenesis in preclinical models [376]. However, a phase I clinical trial (NCT01952249) with MEDI0639 in patients with advanced solid tumors revealed very modest preliminary clinical results, with only 1 out of 20 patients having PR [377]. To our knowledge, no further clinical trials have been completed. In 2016, Xu et al. described a novel DLL4 antibody, MGZ01, which reduced the CD24^−^/CD44^+^ breast cancer CSC population and significantly attenuated tumor growth in MDA-MB-231 xenograft models [378]. Other groups have investigated dual targeting of DLL4 and VEGF with bispecific antibodies such as navicixizumab (OMP-305B83) and ABT-165. Indeed, navicixizumab has been evaluated in phase I clinical trials for patients with solid tumors (NCT02298387) [379]. However, the anti-tumor efficacy of navicixizumab was modest, with 6.1% achieving partial response, 25.8% achieving stable disease, and 57.6% experiencing progressive disease [379]. In a separate phase I trial (NCT03030287), navicixizumab in combination with paclitaxel in patients with ovarian, peritoneal, or fallopian tube cancers resulted in an overall response rate of 43.2% [380]. In preclinical models, ABT-165 induced significant tumor growth inhibition in glioblastoma and colon cancer xenograft models [381]. A phase I trial (NCT03368859) of ABT-165 combined with FOLFIRI for patients with second-line CRC showed modest anti-tumor activity, with 3 of 16 patients displaying a partial response [382].

### 9.3. T-Cell Engagers

Other groups have focused on targeting notch ligands such as the Delta-like ligands (DLL). Tarlatamab (AMG 757) is a DLL3 and CD3 BiTE facilitating T-cell-mediated lysis that displayed excellent anti-tumor activity in preclinical PDX SCLC models [383]. In phase I and II clinical trials for SCLC, tarlatamab displayed promising preliminary results [384,385]. In the phase I trial (NCT03319940), 23 out of 107 patients achieved partial response, while 2 achieved complete response [384]. Similarly, in the phase II trial (NCT05060016), objective response was observed in 40% of SCLC patients [385]. A similar anti-DLL3, anti-CD3, anti-albumin T-cell engager, HPN328, is currently in a phase I/II clinical trial for patients with SCLC and other neuroendocrine cancers (NCT04471727) [386]. The preliminary results indicate moderate tumor control, with 40% of patients experiencing > 30% tumor shrinkage and at least one confirmed partial response.

### 9.4. Antibody Drug Conjugates

Rovalpituzumab terisine (Rova-T) is an antibody–drug conjugate that targets DLL3 [387]. While preclinical models in SCLC were promising, phase II (NCT02674568) and III (NCT03033511) clinical trials failed to reach their efficacy end points [388,389,390].

While targeting Notch signaling pathways, particularly through pan-Notch inhibitors like GSIs, presents a promising therapeutic strategy in cancer treatment, significant challenges remain. The dose-limiting gastrointestinal toxicities associated with GSIs have hindered their clinical translatability despite their demonstrated efficacy in preclinical studies. Furthermore, the limited clinical success observed across various trials highlights the complexity of Notch signaling in cancer and the need for further research to optimize therapeutic approaches. Nevertheless, the diverse array of small molecule inhibitors and antibody-based therapies targeting Notch receptors and ligands offer continued avenues for exploration in the treatment of various malignancies.

## 10. Wnt/β-Catenin

The Wingless-type (Wnt)/β-catenin signaling is a highly conserved pathway governing cell proliferation, survival, migration, differentiation, tissue homeostasis, and self-renewal [391,392]. The Wnt/β-catenin pathway is complex and has been extensively reviewed elsewhere [391,392,393]. There are four major segments in the Wnt/β-catenin pathway, including extracellular signals derived from Wnt proteins, cell membrane receptors such as Frizzled (FZD) and low density lipoprotein receptor-related proteins 5 and 6 (LRP5/6), cytoplasmic β-catenin and disheveled (DVL), and nuclear translocation of β-catenin to modulate expression of target genes through T-cell factor/lymphoid enhancer-binding factor (TCF/LEF) transcription factors [391]. Posttranslational modification of Wnts such as lipid modification through porcupine O-acyltransferase (PORCN) are required for Wnt activation and ligand activity [394]. Active Wnt ligands bind to the FZD-LRP5/6 receptor complex, which, in turn, recruits and activates Dvl to disrupt the β-catenin destruction complex and consists of Axin, adenomatous polyposis coli (APC), glycogen synthase kinase 3β (GSK-3β), and casein kinase 1 (CK1). Cytoplasmic β-catenin is thus stabilized and translocates to the nucleus to interact with TCF/LEF transcription factors to induce the expression of target genes [391,393]. The Wnt/β-catenin pathway is often aberrantly expressed in both cancer and diseases such as atherosclerosis, idiopathic pulmonary fibrosis (IPF), and amyotrophic lateral sclerosis (ALS) [395,396,397]. In cancer, dysregulated Wnt/β-catenin signaling is often due to mutations in one or more components of the Wnt signaling pathway [391]. It is well-appreciated that Wnt/β-catenin signaling is important for maintaining the CSC population in cancer. For example, colon CSCs are characterized by high Wnt expression, with significantly enhanced clonogenicity, as well as tumorigenicity, in limiting dilution assays in mice [398]. In T-ALL Lin−CD3^+^c-KitMid cells, β-catenin overactivation is associated with expansion of the CSC population [399]. Similarly, ALDH1^+^ and CD24^−^/CD44^+^ breast CSCs exhibited increased Wnt/β-catenin signaling activity, and Wnt knockdown or inhibition significantly reduced the tumorsphere formation associated with CSCs [400]. Moreover, Wnt/β-catenin signaling was shown to maintain self-renewal and tumorigenicity in HNSCC CSCs by promoting Oct4 expression [401]. Targeting the Wnt/β-catenin pathway is, thus, an attractive avenue for anti-cancer agents, and numerous agents have been developed to target this pathway from multiple angles (Figure 3, Table 1).

### 10.1. Antibodies

Antibodies targeting various components of the Wnt/β-catenin have been developed and tested against multiple forms of cancer, with many entering clinical trials. Vantictumab (OMP-18R5) is an mAb that binds to FZD1, FZD2, FZD5, FZD7, and FZD8 receptors to inhibit Wnt3a signaling [402]. In colon C28, breast UM-T3, lung Lu24, and pancreatic PN8 xenograft models, vantictumab significantly reduced tumor growth and markedly reduced tumorigenicity in subsequent limiting dilution assays in pancreatic and breast cancer models [402]. Due to its success in preclinical models, vantictumab has progressed to phase I clinical trials; however, its clinical translatability has been limited due to significant bone toxicities associated with Wnt inhibition [403,404,405]. Vantictumab monotherapy in a phase I trial for patients with advanced solid tumors resulted in prolonged SD in 3 out of 18 patients (NCT01345201) [403]. In a 2019 phase I clinical trial for metastatic pancreatic cancer (NCT02005315), partial response was documented in 42% of patients, with an additional 11% achieving stable disease when treated with vantictumab in combination with nab-paclitaxel and gemcitabine [404]. However, the study was terminated due to a failure to reach satisfactory therapeutic index. A similar trial in patients with metastatic HER2-negative breast cancer demonstrated a clinical benefit rate of 68.8% for vantictumab combined with paclitaxel; however, bone toxicity and fragility fractures limited the potential for further development [405]. The Fab F2.A is another antibody modality specific for FZD receptors; binding FZDs 1, 2, 4, 5, 7, and 8 to inhibit Wnt binding [406]. In RNF43-mutant PDAC cells, which rely on Wnt signaling for proliferation, F2.A reduced expression of Wnt target genes NKD1 and AXIN2 and induced G0/G1 cell cycle arrest more effectively than vantictumab [406]. In a similar study, full length IgG human antibodies specific for FZD5, IgG-2919, reduced subcutaneous and orthotopic RNF43-mutant HPAF-II PDAC xenograft growth by up to 74% in immunodeficient mice compared to control [407]. Moreover, IgG-2919 demonstrated more effective tumor control than vantictumab. OTSA101 is a radiolabeled antibody against FZD10 [408]. Indeed, OTSA101 radiolabeling with yttrium-90 (^90^Y-OTSA101), astatine-211 (^211^At-OTSA101), actinium-225 (^225^Ac-OTSA101), or indium-111 (^111^In-OTSA101) has been tested in several preclinical and clinical studies for synovial sarcoma [408,409,410,411]. In synovial sarcoma SYO-1 xenograft models, ^211^At-OTSA101, ^225^Ac-OTSA-101, and ^90^Y-OTSA101 accumulated in the tumor, significantly suppressed tumor growth following one injection, and prolonged survival [408,409]. However, in a phase I clinical trial for patients with synovial sarcoma (NCT01469975), the best recorded response for ^90^Y-OTSA101 was stable disease, with no objective responses observed, and the study was terminated due to slow enrollment [411]. A similar phase I trial (NCT04176016) was halted due to difficulty obtaining raw materials for ^90^Y-OTSA101. Ipafricept (OMP-54F28) is a fusion protein that combines the extracellular ligand-binding domain of FZD8 with an IgG1 Fc domain to inhibit Wnt signaling [412]. In pancreatic and ovarian cancer models, ipafricept combined with paclitaxel decreased tumor growth [413]. Furthermore, ipafricept with sequential paclitaxel reduced tumorigenicity of ovarian OMP-OV40 cells by sevenfold, suggesting depletion of the tumor-initiating CSC population [413]. Multiple phase I trials for ipafricept have been completed for patients with solid, ovarian, or pancreatic tumors [414,415,416]. Ipafricept monotherapy in solid tumors resulted in a best response of stable disease in 26.9% of patients (NCT01608867) [414]. In a subsequent trial for ipafricept in combination with carboplatin and paclitaxel, the overall response rate was 75.7%, with 94.6% of patients receiving clinical benefit (NCT02092363) [415]. Similar results were observed in a trial for pancreatic cancer (NCT02050178), where combination ipafricept with gemcitabine and nab-paclitaxel resulted in a clinical benefit rate of 81%, with 34.6% of patients achieving partial response [416]. The lack of control arm in both studies limits the ability to analyze whether the benefit was derived from ipafricept. Moreover, ipafricept development was halted due to bone toxicities, which presented barriers for commercial viability. Other groups have investigated anti-Wnt1 or R-spondin 3 (RSPO3) antibodies to target the Wnt/β-catenin pathway in cancer. Anti-Wnt1 is a monoclonal antibody specific for Wnt1 that inhibits H460 NSCLC tumor growth in mice [417]. R-spondin ligands like RSPO3 bind to leucine-rich repeat-containing GPCR (Lgr) 4/5/6 to potentiate Wnt signaling [393]. Anti-RSPO3 antibody, rosmantuzumab (OMP-131R10), has been shown to reduce self-renewal capabilities in PDX AML models and decrease the percentage of CSCs; however, clinical trials in advanced colorectal cancer (NCT02482441) were halted due to a lack of clinical efficacy [418,419].

### 10.2. Pharmacologic Inhibition

Numerous small molecule inhibitors of the Wnt/β-catenin pathway have also been developed, with many targeting PORCN. LKG974 (Wnt974) is a small molecule PORCN inhibitor that prevents Wnt activation [420]. In HNSCC xenograft and mouse mammary tumor virus (MMTV)-driven Wnt1 and Wnt3 tumor models, daily injections of LGK974 led to significant tumor regression [420]. LGK974 had an IC50 of 0.3 nM against Wnt-induced gene expression of AXIN2. Remarkably, LGK974 nearly completely abolished established intestinal *RSPO*-fusion tumors in mice in just one week of treatment [421]. Interestingly, LGK974 combined with aspirin more effectively inhibited tumorsphere formation in SW480 and SW742 CRC cell lines compared to oxaliplatin [422]. The results for several phase I clinical trials for LGK974 have recently been published. While LGK974 monotherapy for patients with advanced solid tumors (NCT01351103) did not result in any complete or partial responses, LGK974 in combination with encorafenib and cetuximab in patients with metastatic CRC (NCT0227813) produced marginally better results, with an overall response rate of 10% and disease control rate of 85% [423,424]. Bone toxicities and fractures were a major side effect in both studies. A phase II trial evaluating LGK974 in metastatic HNSCC (NCT02649530) was terminated; but whether this was related to the adverse effects observed in both phase I trials is unknown. ETC-159 is an orally bioavailable PORCN inhibitor that displays an IC50 value of 2.9nM in β-catenin reporter assays [425]. In colon cancer xenografts with *RSPO* fusion genes and *RNF43* mutated pancreatic and ovarian xenografts, ETC-159 markedly reduced tumor growth and prevented tumor recurrence [425]. Histological analysis of the tumors revealed signs of mucinous differentiation with loss of adenocarcinoma. Phase I trials with ETC-159 in advanced solid tumors produced disappointing results, with only 2 out of 16 patients achieving stable disease as best outcome (NCT02521844) [426]. In the subsequent phase Ib trial, ETC-159 was tested in combination with pembrolizumab with slightly improved results (1 partial response, 5 stable disease). All patients have since discontinued, the majority of which due to disease progression, but the study is still considered active (NCT02521844) [427]. C59 is another PORCN inhibitor with a reported IC50 of 74 pmol/L in β-catenin STF reporter assays [428]. In nasopharyngeal carcinoma SUNEI1 and HNE1 xenograft models, C59 significantly reduced tumor growth, completely eliminating HNE1 tumors [429]. Interestingly, C59 also inhibited tumorsphere formation in HNE1 stem-like cells, suggesting that C59 can specifically inhibit stemness in HNE1 cell lines. RXC004 is a PORCN small molecule inhibitor that has progressed to phase II trials. In preclinical models, RXC004 reduced tumor growth and increased differentiation in Wnt-dependent *RNF43*-mutant pancreatic AsPC1 and HPAF-II and *RSPO3*-fusion SNU-1411 colorectal xenografts [430]. Interestingly, RXC004 synergized with anti-PD-1 to increase CD8^+^ T cell infiltration in CRC CT26 tumors. Moreover, RXC004 has been shown to increase survival and reduce tumor growth in *RSPO*-fusion SNU-1411 colorectal xenograft models when used in triplet combination with 5FU, irinotecan and oxaliplatin or doublet combination with 5FU and irinotecan [431]. One phase I study with RXC004 monotherapy in advanced solid cancers demonstrated modest results, with 5 out of 25 patients experiencing stable disease as best response (NCT03447470) [432]. Two phase II studies with RXC004 have commenced, one for patients with *RNF3*-mutant PDAC or biliary tract cancer that was recently completed (NCT04907851) and a second to assess the efficacy of RXC004 in combination with nivolumab in patients with *RSPO* or *RNF43* mutant colorectal cancer (NCT04907539). The latter is actively recruiting. Other small molecule PORCN inhibitors include XNW7201 and CGX1321. XNW7201 is currently in a phase I trial for patients with advanced solid tumors (NCT03901950) and the results have not yet been released. Preliminary results from the phase I trial (NCT02675946) of CGX1321 in combination with pembrolizumab showed promising results for patients with GI tumors bearing *RSPO* or *RNF43* mutations, with 71% of patients achieving stable disease [433].

Other small molecule inhibitors have focused on targeting β-catenin. Tegavivint (BC2059, tegatrabetan) inhibits β-catenin binding to transducing β-like (TBL1), thus preventing β-catenin transcriptional activity [434]. In a panel of desmoid tumor cell lines, tegavivint displayed IC50 values for cell proliferation and viability ranging from 47.79 to 284.7 nM [435]. In an ex vivo model of desmoid tumors, merely 100 nM of tegavivint significantly reduced cell viability. Moreover, tegavivint treatment in in vivo models of AML and multiple myeloma resulted in prolonged survival and decreased tumor burden [434,436]. Interestingly, tegavivint was found to synergize with proteasome inhibitor bortezomib, as well as HDAC inhibitor panobinostat [434,436]. The preliminary results for a phase I clinical trial (NCT03459469) with tegavivint in unresectable desmoid tumors were released in 2022, demonstrating an objective response rate of 25% and a nine-month PFS rate of 79% [437]. There is currently a phase I/II trial for tegavivint recruiting patients with recurrent lymphomas and desmoid tumors (NCT04851119). FH535 is another small molecule β-catenin inhibitor that prevents β-catenin interaction with TCF to inhibit β-catenin-mediated transcriptional activation [438]. In vitro analysis of FH535 revealed that it can potently inhibit TNBC MDA-MB-231 and HCC38 cell growth when cultured in three-dimensional (3D) collagen gels [439]. Moreover, FH535 attenuated the migratory and invasive capacity of both MDA-MB-231 and HCC38 cells at 100 nM concentrations. A later study revealed that FH535 significantly suppressed tumorsphere formation in CD24^−^/CD44^+^ breast CSC [400]. FH535 has also demonstrated anti-CSC activity in pancreatic cancer models, reducing clonogenicity and CD24^−^/CD44^+^ marker expression in vitro, as well as inhibiting metastasis and tumor growth in xenograft models [440,441]. Similarly, FH535 repressed expression of CD24, CD44, and CD133 CSC markers in colon cancer HT29 cells [442]. Interestingly, doxorubicin has also been identified as an inhibitor of β-catenin activity and signaling by preventing the β-catenin/Akt interaction [443]. Low-dose doxorubicin specifically reduced leukemia stem cell (LSC) populations without impacting hematopoietic stem and progenitor cells. Indeed, combining chemotherapy with low-dose doxorubicin significantly improved survival and reduced LSC numbers in the bone marrow of leukemic mice. Moreover, bone marrow transplants from the doxorubicin and chemotherapy treated leukemic mice displayed significantly reduced tumorigenicity when implanted into secondary recipients, further supporting the notion that low-dose doxorubicin directly targets LSCs [443]. FOG-001 is a first-in-class direct β-catenin inhibitor that blocks β-catenin interaction with TCF and is currently recruiting for phase I trials in advanced metastatic tumors (NCT05919264).

Several papers have also been published using FZD, DVL, and β-catenin transcription co-activator cAMP-responsive element binding (CREB)-binding protein (CBP) inhibitors to target Wnt/β-catenin. Carbamazepine is a well-known antiepileptic drug that was recently found to bind to FZD8 to inhibit Wnt signaling; however, it has yet to be fully explored as an anti-cancer agent [444]. FJ9 is a small molecule inhibitor that disrupts FZD7 binding to DVL, thus repressing Wnt-mediated signaling [445]. In NSCLC H460 and H1703 cell lines, FJ9 caused significant apoptosis. Moreover, in NSCLC H460 xenograft models, FJ9 treatment for 16 days reduced tumor volume [445]. Selective FZD7 peptide inhibitors have also been developed, with dFz7-21 having a reported IC50 of 100 nM against Wnt/β-catenin signaling [446]. While dFz7-21 reduced the LRG5-GFP^+^ intestinal stem cell population in organoids, it has not yet been tested as an anti-cancer agent [446]. Interestingly, niclosamide, which is an antihelminth compound, was found to specifically downregulate Dvl2 expression, resulting in antiproliferative effects in CRC HCT116 cells and suppression of tumor growth in HCT116 xenograft models [447]. Niclosamide nanosuspension (nano-NI) has improved the anti-cancer profile of niclosamide, showing improved inhibitory effects on ovarian tumorsphere formation and significantly reducing ovarian CP70 and SKOV3 tumor growth in mice [448]. One phase I trial (NCT02532114) for niclosamide in combination with enzalutamide for castration-resistant prostate cancer revealed a lack of clinical benefit, and the trial was terminated [449]. A phase II trial (NCT02519582) investigating the efficacy of niclosamide in CRC commenced in 2015; however, its status is currently unknown. Screening of a synthetic chemical library unveiled an Axin-stabilizing small molecule, IWR-3, that induces β-catenin destruction. Indeed, IWR-3 was shown to inhibit Wnt/β-catenin signaling in DLD-1 CRC and DU145 prostate cancer cells; however, its efficacy as an anti-tumor agent has yet to be investigated [450]. PRI-724 is a first-in-class CBP inhibitor that has progressed to clinical trials in PDAC (NCT01764477), advanced solid tumors (NCT01302405); however, the best response in both trials was stable disease [451,452].

To conclude, the Wnt/β-catenin signaling pathway is a fundamental regulatory mechanism and is especially relevant in the maintenance of CSCs. Targeting components of the Wnt/β-catenin pathway, such as receptors, ligands, and downstream effectors, has emerged as a promising strategy for cancer therapy. Despite challenges such as bone toxicities and limited clinical efficacy observed in some trials, the ongoing research and clinical investigations continue to explore the therapeutic potential of targeting the Wnt/β-catenin pathway in cancer treatment. Further understanding of the intricate regulation of this pathway and the development of novel targeted therapies hold the potential to significantly impact the management of various cancers in the future.

## 11. SOX2

SOX2 is a member of the SRY-related high mobility group (HMG) box family of transcription factors and is essential in embryonic development. SOX2 is a master transcription factor in the early embryo and is required for the regulation of other key developmental genes like ZFP148 and KIT. Deletion of the SOX2 gene is embryonic lethal SOX2 [453,454]. Functional SOX2 is required to maintain the pluripotential stemness required for further differentiation and growth of extraembryonic ectoderm tissues from the epiblast [455]. Furthermore, cells lacking SOX2 lose their pluripotential identity and differentiate into another cell type. In their seminal paper, Takahashi and Yamanaka identified that SOX2 is a key gene that when overexpressed in somatic differentiated fibroblast cells will induce a stem cell phenotype [456]. SOX2 plays key roles in the normal physiology of adult tissues. Arnold et al. determined that SOX2 is a bona fide stem cell marker in adult epithelial tissues, and SOX2^+^ cells are capable of self-renewal and give rise to differentiated progeny [457]. SOX2 is expressed in stem-cell-niches of adult tissues including the brain, lens, glandular compartment of the stomach, and lung [453].

The embryonic stem-cell-like gene signature is associated with more aggressive and poorly differentiated tumors, including breast, lung, skin, esophageal, glioblastoma, and bladder cancer [453,455,458]. SOX2 is a key driver of CSC-related protumor functions in many cancer types [459,460]. The overexpression of SOX2 resulted in increased proliferation, EMT signature, migration, and invasion of lung cancer cells [461]. In breast cancer, the patients whose tumors were enriched with embryonic stem cell genes, like SOX2, Nanog, and OCT4, had significantly worse overall survival compared to those with a non-stem-cell genotype [453]. In nasopharyngeal carcinoma CSCs, SOX2 expression was associated with high risk of tumor progression and metastasis [462]. SOX2 expression is significantly higher in squamous cell carcinoma (SCC) CD34^+^ tumor initiating cells compared to CD34^−^ cells [463]. In colorectal cancer, SOX2 was shown to promote stemness, chemoresistance, and EMT [464]. Thus, the role of SOX2 as an integral stemness-promoting factor in CSCs has made it an attractive target for anti-cancer therapies (summarized in Table 1).

### 11.1. ZF-ATFs and PIPs

SOX2 lacks a binding pocket, so the development of direct SOX2 inhibitors is not yet available. Some groups have attempted to inhibit SOX2 at the transcriptional level. Stolzenburg et al. used a zinc finger artificial transcription factor (ZF-ATF) delivered by retrovirus to selectively suppress SOX2 in breast cancer cell lines [465]. The ATFs bind target gene promoter regions on the DNA, silencing the gene. Induction of the SOX2-specific ZF-ATFs, ZF-522SKD or ZF-598SKD, in breast MCF-7 and MDA-MB-435 cells resulted in decreased SOX2 mRNA and protein expression, as well as decreased cell viability and colony formation in vitro. Moreover, ZF-598SKD significantly reduced tumor growth in MCF-7 xenograft models [465]. ATF/SOX2 is another ZF-ATF against SOX2, which has been tested in mouse models of lung squamous cell carcinoma [466]. EBC2 lung cancer cells transfected with ATF/SOX2 failed to grow tumors in nude mice [466]. Other groups have used DNA-binding inhibitors to prevent SOX2 transcriptional activation [467]. Though it has not yet been tested as an anti-cancer agent, PIP-S2 successfully inhibited SOX2-DNA binding by targeting the SOX2 DNA-binding sequence, thus triggering the differentiation of pluripotent stem cells [467]. While the therapeutic translatability of this method is unclear, PIP-S2 appears to abrogate SOX2-mediated stemness. Collectively, these results demonstrate a template for upstream targeting of SOX2; however, further study is required to confirm their validity.

### 11.2. Peptides

Direct peptide antagonists for SOX2 have also been developed. Indeed, a SOX2-targeting peptide, peptide 42 (sP42), significantly reduced esophageal squamous cell carcinoma (ESCC) KYSE450 and TE-1 proliferation, as well as migration and invasion [458]. Furthermore, treating xenograft models of KYSE450 ESCC cells with sP42 resulted in significant tumor growth inhibition in mice and decreased metastatic foci in zebrafish [458]. Interestingly, sP42 also promoted terminal differentiation in KYSE450 tumors. More recently, a SOX2-interference peptide (SOX2-iPEP) consisting of a truncated 24 amino acid sequence from the C-terminus of SOX2, which includes a potential nuclear localization sequence and binding sites for SOX2 interactions with DNA and transcription factors, led to increased apoptotic death in MCF-7 and PA-1 cells [468].

### 11.3. Immunomodulatory Peptides and Vaccines

Other groups have investigated immunomodulatory peptides against SOX2. Amino-acid sequence screening of SOX2 revealed three alleles predicted to bind to human leukocyte antigen (HLA)-A*0201 to promote CD8^+^ T cell activation [469]. CD8^+^ T cells stimulated with SOX2 peptide 60030 efficiently lysed SOX2-expressing glioma cell lines. In a separate study, oligodendroglioma tumor-bearing mice “vaccinated” with SOX2 peptides in combination with temozolomide lived twice as long as both untreated mice and mice treated with temozolomide [470]. Recently, several phase I and II clinical trials (NCT05242965, NCT05455658, NCT02157051) have commenced for STEMVAC, a DNA plasmid-based vaccine that targets multiple cancer antigens, including SOX2. These studies are currently ongoing, and the results have not yet been published.

### 11.4. Pharmacologic Inhibition

The majority of pharmacologic inhibitors targeting SOX2-dependent cancer growth do not target SOX2 directly but inhibit SOX2-associated signaling pathways and molecules. SOX2 is involved in many cancer pathways, and the PI3K/Akt/mTOR pathway has been shown to positively regulate SOX2 expression [471]. Rapamycin is a specific mTOR complex 1 inhibitor that recently gained FDA approval for advanced perivascular epithelioid cell tumors [472]. Interestingly, rapamycin decreased SOX2 mRNA expression in glioma U251 and U373 cell lines and inhibited U251 glioma tumor growth in mice [473]. Other groups have focused on targeting Akt, as Akt has been shown to support SOX2 overexpression by preventing its ubiquitination and degradation by the proteosome in ESCC [474]. Indeed, a study of ESCC determined that SOX2 levels do not correlate directly to gene copy number but instead to cellular Akt levels [474]. Indeed, MK2206, which is an Akt inhibitor, reduced SOX2 protein levels and suppressed tumor spheroid formation in ESCC K450 cells [474]. MK2206 has been investigated in a phase II trial (NCT01277757) for patients with advanced breast cancer; however, it displayed very little clinical efficacy [475]. Similarly, Qin et al. showed that Akt inhibitor DC120 was able to indirectly downregulate SOX2 protein and mRNA levels in nasopharyngeal carcinoma CSCs by upregulating tumor suppressor p27 [476]. In a preclinical model of nasopharyngeal carcinoma, DC120 significantly reduced tumor volume and CSC in both primary and secondary xenograft tumors, which they attributed to a decrease in SOX2 expression [476].

SOX2 is also a downstream target of growth receptor pathways. Indeed, SOX2 expression is stimulated by fibroblast growth factor receptor 1 (FGFR1) activation in lung cancer [461]. FGFR1-amplification in lung cancer cell lines H1581 and DMS1144 was shown to promote EMT, migration, and invasion, which were attributed to SOX2 expression [461]. Mechanistically, FGFR1 promotes ERK1/2 phosphorylation, which promotes the expression of SOX2. AZD4547 is a FGFR1 inhibitor that markedly suppressed SOX2 expression and tumor growth in orthotopic and subcutaneous H1581 and DMS1144 lung cancer xenograft models [461]. More recently, AZD4547 has been investigated in phase II clinical trials (NCT01791985) for endocrine-resistant breast cancer in combination with anastrozole or letrozole; however, only 10% of patients exhibited partial response [477]. SOX2 expression is also modulated by epidermal growth factor receptor (EGFR)/Src/Akt signaling pathways. Indeed, inhibition of EFGR/Src/Akt signaling with EFGR inhibitor gefitinib, Src inhibitor dasatinib, and Akt inhibitor LY294002 specifically decreased SOX2 expression in NSCLC H1650SP-Adh CSCs [478]. Moreover, EFGR inhibitors gefitinib and erlotinib both decreased the proportions of CSCs in NSCLC and reduced spheroid formation and size [478]. Currently, gefitinib and erlotinib, as well as dasatinib, are FDA-approved for treatment of NSCLC and some forms of leukemia, respectively. Interestingly, gentian violet, which is an antimycotic and antibacterial agent, was found to reduce EFGR activation and STAT2 phosphorylation, thus preventing STAT2 binding to the SOX2 promoter [479]. Indeed, gentian violet treatment impaired melanoma CSCs by inhibiting SOX2 expression, resulting in reduced sphere-forming capacity, with enhanced effects seen on second generation spheroids, in a dose-dependent manner. The effect was abolished with SOX2-overexpression [479]. Together, this is indicative that growth factor receptor inhibitors decrease stem cell capabilities in multiple SOX2-driven cancer types.

To identify other indirect targets of SOX2, Kim et al. studied the interactome and found that SOX2 interacts with transcription factors, chromatin remodelers and modifiers, cyclin-dependent kinases, and others [480]. They identified EP300 as a key mediator of SOX2 expression, and CBP30, which is an EP300 inhibitor, reduced MGH7 and ChaGoK1 lung cancer cell growth in vitro [480]. Interestingly, CBP130 synergized with PI3K inhibitor, BKM120, to further decrease growth [480]. Others have identified pevonedistat (MLN4924), a neddylation inhibitor, as a potent indirect SOX2 inhibitor [481]. Mechanistically, pevonedistat inactivates the E3 ligase F-Box and WD repeat domain containing 2 (FBXW2), preventing Msh homeobox 2 (MSX2) ubiquitylation and degradation. MSX2 acts as a SOX2 transcription repressor, and pevonedistat-mediated MSX2 stabilization results in decreased SOX2 mRNA and protein expression [481]. Pevonedistat sensitized the tamoxifen-resistant MCF-7 cells, resulting in significant tumor growth control. Pevonedistat in combination with various chemotherapy agents has been tested in several clinical trials (NCT03268954, NCT03323034, NCT03770260, NCT03965689), with four patients achieving complete response in a trial for AML [482,483]. Identifying the coactivators of SOX2 may be an effective method of targeting the tumorigenic properties it bestows.

Other groups have targeted factors impacting SOX2 protein stability. In a preclinical model of nasopharyngeal carcinoma, XIAP maintained stemness by stabilizing the SOX2 protein, extending its half-life by inhibiting autophagic degradation [462]. The drug APG-1387 is a second mitochondria-derived activator of caspases (SMAC) mimetic that directly binds to X chromosome-linked inhibitor of apoptosis protein (XIAP), leading to the induction of apoptosis [484]. APG-1387 has shown anti-CSC activity, potently reducing CSC populations in nasopharyngeal carcinoma S-18 cells [462]. In primary S-18 xenograft mouse models, APG-1387 reduced tumor growth compared to control and synergized with 5FU or cisplatin, resulting in a significant reduction in tumor growth. Moreover, secondary xenografts displayed significantly slower growth following treatment with APG-1387 alone or in combination with 5-FU or cisplatin, indicating at least a partial eradication of CSCs.

The targeting of epigenetic mechanisms to repress SOX2 has also been investigated as an anti-cancer mechanism. Histone deacetylase 11 (HDAC11) is a chromatin remodeler that has recently been implicated as an inducer of SOX2 expression in NSCLC [485]. Selective HDAC11 inhibitors, FT234 and FT895, reduced SOX2 expression in H1650 and A549 NSCLC cells and suppressed stemness properties, evidenced by reduced self-renewal and tumorsphere formation [485]. Similarly, lysine-specific demethylase 1 (LSD1) demethylates lysine residues on histone H3 and is an essential epigenetic regulator of pluripotency [486,487]. Moreover, LSD1 is required for SOX2 expression in lung cancer H520, A2780, and T47D cells and is positively correlated with SOX2 expression in lung squamous cell carcinoma patient samples [486]. Use of an LSD1 inhibitor, CBB1007, enhanced repressive histone H3 methylation at the SOX2 enhancer, resulting in the reduced binding of SOX2 to its target genes [486]. Iadademstat (ORY-1001) is a highly potent LSD1 inhibitor that suppressed enhancer-driven activation of SOX2 in BT-474 breast cancer cells [487]. Moreover, iadademstat caused a dose-dependent decrease in tumor spheroid formation in MDA-MB-436 TNBC cells with no effect on viability assays, demonstrating that the drug effectively and specifically targets stemness potential without off-target effects [487]. Phase I trials for iadademstat have been completed in AML (EUDRACT 2013-002447-29), with an objective response rate of 36% [488]. Other clinical trials for iadademstat in combination with paclitaxel in SCLC or gilteritinib in AML are currently recruiting (NCT05420636, NCT05546580).

SOX2 is a pivotal player in both normal physiological processes and cancer progression, particularly as a driver of CSC properties across various cancer types. It has been clearly demonstrated that targeting SOX2 is a potent method of inhibiting CSCs and their tumorigenic potential. While direct inhibition of SOX2 has posed challenges due to its lack of a binding pocket, alternative strategies targeting its transcriptional regulation, protein stability, and epigenetic modulation have shown promise in preclinical and clinical studies. Further research and clinical trials are warranted to validate and optimize these approaches for clinical use in cancer therapy.

## 12. Hedgehog

The hedgehog (HH) pathway is a signal transduction pathway that confers gene expression changes, activated by HH ligands. The pathway was first discovered in 1980 in drosophila melanogaster and determines segment polarity [489]. There are three mammalian HH pathway ligands sonic hedgehog (SHH), Indian hedgehog (IHH), and Desert Hedgehog (DHH). SHH and IHH are present in most tissues, and SHH is the most extensively studied [490]. DHH is present in gonads [491]. HH signaling is essential in determining the cell fate, cell patterns and signals between cell growth and differentiation [490]. The canonical HH pathway involves two transmembrane receptors smoothened (SMO), a G-protein coupled receptor and patched 1 (PTCH). Upon ligand binding to PTCH, a SMO repressor, SMO is accumulated in primary cilium and downstream processing and activation of glioma-associated oncogene (GLI) family transcription factors are initiated [492]. Further, GLI proteins activate transcription of its targets. The HH pathway is silenced in adult tissue unless aberrant activation in case of certain cancers. Several cancers have aberrant HH pathway activation. It is well appreciated that HH signaling is important for the maintenance of CSCs across multiple cancer types [489,490]. Indeed, HH signaling components are overexpressed in CD24^−^/CD44^+^ breast CSCs [491]. Similarly, HH signaling was found to be essential for maintenance of CSCs in chronic myelogenous leukemia (CML), as SMO^−/−^ cells displayed reduced tumorigenicity in CML mouse models [493]. Isolated ALDH^+^ melanoma CSCs display enhanced tumorigenicity and self-renewal capabilities, effects which are abrogated upon HH signaling inhibition [492]. In pancreatic cancer, HH signaling was found to regulate pancreatic CD133^+^ CSC properties such as tumorsphere formation [494]. Thus, targeting the components of HH signaling present attractive anti-cancer targets (Figure 4, Table 1). 

### 12.1. Pharmacologic Inhibitors

With respect to inhibition of the HH pathway, most drugs inhibit the HH target SMO, with several receiving FDA approval as anti-cancer agents [495]. Vismodegib (GDC-0449) is a small molecule antagonist of SMO and was the first FDA approved HH pathway inhibitor for cancer [496]. In preclinical studies on pancreatic cancer CSCs, vismodegib suppressed primary and secondary spheroid formation, indicating that it can directly inhibit CSC characteristics [497]. Keyword searches in the ClinicalTrials.gov database showed 85 different studies in several solid tumors studying vismodegib as a monotherapy or in combination, for which results have been extensively reviewed elsewhere [496]. Vismodegib monotherapy in phase I and II trials showed promise in basal cell carcinoma (BCC) patients with high overall response rates (NCT00607724, NCT00833417) [498,499]. However, clinical trials in other cancer types including refractory gastric cancer (NCT03052478) and ovarian cancer (NCT00739661) showed less promising results [500,501]. In a phase II clinical trial for metastatic PDAC (NCT01088815), vismodegib in combination with gemcitabine and paclitaxel showed better response with compared to monotherapy, with a 40% objective response rate [502]. Sonidegib (LDE225) is the second drug following vismodegib to be approved by FDA for BCC in 2015 [503]. Irvine et al., 2016 showed sonidegib treatment by itself or in combination with nilotinib (tyrosine kinase inhibitor against BCR-ABL oncogene of CML) reduced the self-renewal capacity of chronic phase (CP) CML in vitro and when engrafted to NSG mice [504]. Further, sonidegib reduces multi-drug resistance p glycoproteins ABCB1 and ABCG2, while their gene levels are not altered in NSCLC, this increases cytotoxicity of other drugs like mitotraxane and daunorubicin [505]. Clinical trials using sonidegib have shown high success rates in some cancers [503]. Indeed, in BCC, sonidegib monotherapy showed 56% ORR (NCT01327053) while combination therapy using etoposide and cisplatin on extensive SCLC showed 79% partial response (NCT01579929) [506,507]. In a small phase I trial, sonidegib in combination with docetaxel led to complete response in one patient (NCT02027376) [508]. Sonidegib has also been tested in clinical trials for medulloblastoma (NCT01125800), and CML (NCT01456676) with varying degrees of efficacy [503]. Glasdegib (PF-04449913) is another SMO inhibitor with FDA-approval that has been used in clinical trials, mainly for treating acute myeloid leukemia [509,510]. In the phase II BRIGHT AML 1003 clinical trial (NCT01546038), glasdegib improved the efficacy of low dose cytarabine (LDAC) treatment in myeloid leukaemia patients ineligible for intensive chemotherapy, improving complete remission from 2.6% in LDAC treated patients to 19.2% patients in combination therapy [511]. However, in the phase 3 BRIGHT AML 1019 study (NCT03416179) on patients with untreated AML, glasdegib showed no benefit with intensive treatment of cytarabine or danorubicin, and the efficacy endpoints of the study were not met [512].

Cyclopamine is a steroidal alkaloid that directly binds to the heptahelical bundle of SMO, however it displayed teratogenic properties which limited its clinical translatability [513,514]. Second generation derivatives of cyclopamine have yielded several HH-inhibiting compounds including vismodegib (described above), IPI-269609, and saridegib (IPI-926). IPI-269609 reduced Gli1 mRNA, as well as the ALDH^+^ pancreatic E3LZ10.7 cell population, and impaired colony formation [515]. Later modifications of IPI-269609 yielded saridegib, a potent SMO inhibitor which completely abrogated tumor regrowth in B837Tx medulloblastoma xenograft models following treatment with saridegib, indicating complete elimination of cancer cells [516]. In chondrosarcoma xenograft tumors, saridegib inhibited GLI1 expression and significantly reduced tumor volume in mice [517]. Moreover, histological analysis revealed increased calcification in the tumors, suggesting that IPI-926 may drive chondrosarcoma differentiation [517]. Multiple clinical trials have been completed for saridegib with relatively modest results [495]. In a small pilot study for HNSCC (NCT01255800), saridegib in combination with cetuximab yielded one partial response, and a median PFS of 77 days [518]. IPI-926 showed some anti-tumor activity in a separate phase I clinical trial for HNSCC, with two complete responses and six partial responses in patients with BCC [519]. FOLFIRINOX (5FU, leucovirin, irinotecan, oxaliplatin) in combination with IPI-926 displayed an unconfirmed objective response rate of 66.7% in patients with pancreatic cancer (NCT01383538) [520]. A double-blind placebo-controlled phase II trial for chondrosarcoma, revealed that IPI-926 therapy had no effect on PFS, or overall survival compared to placebo (NCT01310816) [521]. 

Other SMO inhibitors include BMS-833923 (XL139), taladegib (LY2940680, ENV-101), and Hh003. BMS-833923 was tested in esophageal adenocarcinoma cell lines and was shown to decreased cell proliferation and increase apoptosis [522]. In ovarian cancer cell lines, BMS-833923 synergized with carboplatin to induce cell death [523]. Moreover, BMS-833923 was shown to significantly decrease the quiescent CD34^+^/CD38^−^ LSC population in AML in vitro [524]. BMS-833923 plus dasatinib has also been investigated in a phase I clinical trial (NCT01218477) for patients with CML; however, it displayed very limited clinical efficacy [525]. Taladegib was shown to potently inhibit HH signaling in medulloblastoma cell lines, and significantly prolonged survival, and inhibited tumor growth of Ptch^+/−^ p53^−/−^ transgenic mice which spontaneously develop medulloblastoma [526]. Multiple clinical trials for taladegib have been completed, (NCT01919398, NCT02784795). In one phase I trial, taladegib displayed a 46.8% response rate in patients with BCC, five of which were complete response (NCT01226485) [527]. There is currently one active phase II study evaluating taladegib advanced solid tumors with PTCH1 loss of function mutations (NCT05199584). While Hh003 has not yet been tested in clinical trials, it was shown to inhibit the colony formation of colorectal HCT116 and pancreatic Panc-1 cells more effectively than vismodegib [528]. Furthermore, Hh003 significantly suppressed HCT116 tumor growth in mice compared to control and vismodegib treatment [528].

Itraconazole is an azole-based FDA-approved antifungal drug, which has been repurposed in preclinical and clinical trials as an anti-cancer agent [529]. The exact mechanism by which itraconazole inhibits HH signaling is not entirely clear, with some groups suggesting SMO inhibition, and others implicating inhibition of Gli1 and HH expression [530,531]. Itraconazole treatment in mouse xenograft models of medulloblastoma, colorectal, and breast cancer impaired tumor growth [530,531,532]. In all these studies inhibition of HH signaling and reduction of GLI levels were observed. Retrospective analysis of patients with TNBC, metastatic pancreatic cancer or biliary tract cancer who received chemotherapy followed by itraconazole revealed promising results, with median overall rates ranging 47–62% [533,534,535]. In a phase II clinical trial (NCT00769600) investigating itraconazole in combination with pemetrexed in NSCLC, itraconazole improved significantly PFS compared to pemetrexed alone, with overall survival increasing from 8 months to 32 months [536]. High dose itraconazole also increased prostate-specific antigen progression-free survival rates to 48% at 24 weeks in a phase II clinical trial for castration resistant prostate cancer [537].

Arsenic based drugs (arsenicals) inhibit GLI2, activator of HH dependent transcription [495]. Three different arsenicals, Sodium arsenite, Arsenic trioxide (ATO) and phenylarsine oxide showed reduction of HH signalling in medulloblastoma cells, resulting in their death [538]. ATO binds to GLI family transcription factors to inhibit their ciliary localization and further nuclear translocation in medulloblastoma and in promyelocytic leukemia [538,539]. In SCLC CSCs, ATO significantly inhibited GLI1 expression, resulting in reduced clonogenicity and spheroid formation, as well as tumorigenicity SCLC xenograft models [540]. Similar effects were observed when Ewing Sarcoma cells were treated with ATO, reducing GLI1 levels, resulting in decreased tumor growth [541]. Moreover, in pancreatic cancer xenografts, ATO in combination with gemcitabine reduced CD24^−^/CD44^+^ and ALDH1A1^+^ CSC populations and inhibited tumor growth [542]. 

GANT61 is a small molecule which inhibits GLI1/2 DNA binding [543]. GANT61 increased apoptosis in biliary tract cancer cell lines in vitro [544]. In melanoma tumorspheres with CSC characteristics and high ALDH expression, GANT61 significantly reduced tumorsphere formation, suggesting targeting of the CSC population [492]. Similarly, GANT61 abolished tumorsphere formation in CD133^+^ pancreatic cancer cells and undifferentiated HLE and HLF hepatocellular carcinoma cells, further implicating GANT61 as a direct CSC inhibitor [494,545]. GANT61 has not yet progressed to clinical trials, likely due to relatively high doses required to see anti-CSC effects.

### 12.2. Antibodies

Only a few antibodies targeting the HH signaling pathway have been explored for anti-cancer activity. MEDI-5304 is a human antibody with specificity for both SHH and IHH [546]. Interestingly, despite demonstrating anti-tumor activity in HT-29 colon xenografts at a dose of 1 mg/kg, MEDI-5304 did not reduce tumor growth in pancreatic P479 subcutaneous xenografts, or the frequency of CSCs in the pancreatic tumor explants [546]. Whether MEDI-5304 reduced CSC frequency in CRC was not examined. The authors also tested the efficacy of 5E1, which is a mouse mAb specific for SHH [496]. While MEDI-5306 displayed high SHH binding affinity than 5E1, both antibodies displayed similar anti-tumor activity in CRC xenograft models [546]. In pancreatic PDX tumor models, 5E1 displayed moderate anti-cancer activity, preventing tumor growth and even facilitating tumor regression [547]. In a mouse model of breast cancer, 5E1 has demonstrated the ability to reduce both tumor size and the occurrence of liver and pancreatic metastases [548]. Similarly, 5E1 decreased primary HCT8 colon xenograft tumor size in nude mice, as well as lung metastasis in chorioallantoic membrane chick embryo models [549]. α-Ptch1 is a novel anti-PTCH1 antibody which reportedly reduced proliferation of pancreatic cancer cells; however, the data on this antibody is limited [550]. 

Targeting the HH signaling pathway shows promise as a therapeutic approach for various cancers; however, there are challenges that need to be addressed. Pharmacological inhibitors such as vismodegib, sonidegib, and glasdegib have demonstrated efficacy in clinical trials, but one limiting factor is the potential development of resistance due to acquired mutations in the SMO receptor. Despite this challenge, second-generation inhibitors and repurposed drugs offer alternatives for overcoming resistance and improving treatment outcomes. Ongoing research into novel inhibitors and combination therapies, along with a deeper understanding of the mechanisms underlying HH signaling, are essential for maximizing the therapeutic potential of targeting this pathway in cancer treatment. Additionally, efforts to develop antibody-based therapies targeting HH signaling components may provide new avenues for intervention in HH pathway-driven malignancies.

## 13. Conclusions

In this review, we have presented a comprehensive summary of the therapeutic targets associated with CSCs, specifically focusing on the CSC markers and pathways that are most prevalent across multiple cancer types. The studies on CSCs have highlighted pathways and factors that are important in tumor progression and recurrence and thus present as attractive targets, regardless of their association with CSCs. Our analysis highlights the diverse array of molecular targets associated with CSC stemness, including ALDHs, CD44, EpCAM, CXCR4, Sox2, CD55, CD133, and signaling pathways such as Wnt/β-catenin, Notch, and Sonic Hedgehog. Indeed, the therapeutic landscape for therapies that target these cancer-promoting factors that have been ascribed to CSCs is expansive and diverse. From pharmacologic small molecule inhibitors and antibody–drug conjugates to immunomodulatory agents such as antibodies, CAR-T cells, CAR NK cells, BiTEs, immunotoxins, and vaccines, there exist a plethora of strategies that could be used to target CSCs. It is important to note that although these factors are associated with CSCs, many of the studies discussed were not specifically aimed at targeting CSCs as a read out of efficacy and often assessed only if tumor growth was impeded. They may have found the unintended beneficial side-effect of reducing CSCs if this had been assessed.

Despite the remarkable progress made in developing therapies against factors and markers associated with CSCs, several challenges persist in translating these findings into clinically effective therapies. Notably, many of the molecular targets and pathways associated with CSCs are also expressed in normal physiological processes and other stem cell populations. This presents a significant hurdle, as on-target effects of therapeutic interventions could inadvertently impact normal tissue function, leading to potential toxicity and adverse effects. Furthermore, while preclinical studies have demonstrated the efficacy of these novel agents in inhibiting tumor growth and metastasis, the translation of these findings into successful clinical outcomes has been limited. While clinical studies have provided invaluable insights into the efficacy of targeting factors associated with CSC therapies, none have yet been approved as specific anti-CSC treatments. Instead, the therapeutic agents that have gained regulatory approval function as broad-spectrum anti-cancer agents, coincidentally targeting molecules and pathways that are elevated in CSCs. This underscores the complexity of CSC biology and the challenges associated with developing therapies that selectively eradicate CSCs while sparing normal cells.

Moving forward, addressing the specificity of the targeted therapies will be paramount to mitigate off-target effects and enhance therapeutic efficacy. Additionally, continued research efforts are needed to identify novel CSC-specific vulnerabilities and refine therapeutic strategies that exploit these targets. By leveraging the collective knowledge gained from preclinical investigations and clinical trials, we can advance the development of more precise and effective targeted therapies that in addition to inhibiting tumor progression, target CSCs and ultimately improve the long-term outcomes for cancer patients.

**Table 1 ijms-25-04102-t001:** Drugs targeting CSC markers and pathways with types of cancer tested in and clinical status.

CSC Target	Drug	Condition	Clinical Status	Reference/Trial Number (if Applicable)
ALDH1A1	NCT-501	HNSCC, pancreatic, CRC	Preclinical	[42,43,44]
	CM37	Ovarian	Preclinical	[45]
	Compound 974	Ovarian	Preclinical	[46]
ALDH1A3	GA11	Glioma	Preclinical	[50]
	MF-7	Breast	Preclinical	[12]
	NR6	Glioblastoma, CRC	Preclinical	[51]
	MCI-INI-3	Glioma	Preclinical	[52]
	YD1701	CRC	Preclinical	[53]
ALDH2	CVT-10216	CRC	Preclinical	[13]
	Daidzin	CRC	Preclinical	[13]
ALDH3A1	Dyclonine	SCC, gastric	Preclinical	[60]
	CB57	Lung, glioblastoma	Preclinical	[61]
	CB29	Glioblastoma	Preclinical	[62]
	EN40	Lung	Preclinical	[64]
Pan-ALDH	DEAB	Melanoma, pancreatic	Preclinical	[66,67,68]
	NanoKS100	Melanoma	Preclinical	[69]
	DIMATE	AML	Preclinical	[70,71]
	637A	Ovarian	Preclinical	[72]
	Citral	Breast	Preclinical	[73]
	Disulfiram	Breast, lung	Preclinical	[73,74,75]
EpCAM	EpAb2-6	CRC, SCC, pancreatic, lung,	Preclinical	[90,91]
	Adecatumumab (MT201)	Ovarian, breast, prostate	Phase II	[93,94,95]
	AM-928	Solid tumors	Phase I	NCT05687682
	Solitomab (MT110)	Pancreatic, ovarian, solid tumors,	Phase I	[98,99,100]
	Catumaxomab	Malignant ascites	Approval withdrawn in European Union	[101]
	EpCAM-CD3 hFc mRNA-LNP	Ovarian	Preclinical	[102]
	Anti-EpCAM CAR-T cells	Ovarian, prostate, lung, gastric, pancreatic	Preclinical	[103,104,105,106]
	IMC001	GI tumors	Phase II	[108] NCT05028933, NCT04196465
	VB6-845	Ovarian, breast, SCLC, CRC, SCC	Phase I	[109,110,111] NCT00481936
	VB4-845	HNSCC, bladder carcinoma	Phase III	[113,114,115,116] NCT04859751
	SyntOFF	Breast	Preclinical	[119]
	chiHEA125-Ama	Pancreatic	Preclinical	[120]
	Anti-EpCAM siRNA	Breast, retinoblastoma	Preclinical	[123,124]
CD44	Bivatuzumab (BIWA-4)	HNSCC	Phase I	[133] NCT02254018
	H4C4	Pancreatic	Preclinical	[134]
	IM7	Breast	Preclinical	[135]
	RG7356 (RO5429083)	Leukemia, HNSCC, solid tumors	Phase I	[136,137,138,139,140] NCT01358903, NCT01641250
	Anti-CD44-IR700 NIR-PIT	SCC, CRC, lung	Preclinical	[142,143,144]
	rhPRG4	Breast	Preclinical	[145,146]
	Apt#7	Breast	Preclinical	[149]
	CD44-EpCAM aptamer	Ovarian	Preclinical	[150]
	ASO 4401	HCC	Preclinical	[152]
	THIQ	HNSCC	Preclinical	[154,155]
	JE22-NP	Breast	Preclinical	[156]
	Verbascoside	Glioblastoma	Preclinical	[157]
	A6 (SPL-108)	Prostate, multiple myeloma, breast, ovarian, leukemia	Phase II	[158,159,160,161,162,163,164,165] NCT00939809, NCT02046928
CD55	Anti-CD55	CRC	Preclinical	[172]
	MB55	Lymphoma, leukemia	Preclinical	[173]
	CD55 NAb	Neuroblastoma	Preclinical	[168]
	177Lu-anti-CD55	Lung	Preclinical	[174]
	GB262	Pancreatic	Preclinical	[175]
	105AD7	CRC, osteosarcoma	Phase II	[176,177,178,179]
	PAT-SC1	Gastric	Phase I	[180]
	CD55sp	Cervical	Preclinical	[181]
	C-PC/CMC-CD55	Cervical	Preclinical	[182]
	AWT-489	CRC	Preclinical	[184]
	CRISPR/cas9	Cervical	Preclinical	[171]
	siRNA	Breast, ovarian, lung	Preclinical	[185]
CXCR4	Ulocuplumab (BMS-936564, MDX-1338)	Leukemia, lymphoma, multiple myeloma, breast, Waldenström macroglobulinemia	Phase II	[199,200,201,202,203] NCT01120457, NCT02472977, NCT02305563, NCT03225716, NCT01359657, NCT02666209
	PF-06747143	Leukemia	Preclinical	[204,205,206]
	12G5	Endometrial, osteosarcoma	Preclinical	[207,208]
	ALX-0651	N/A	Phase I	[209] NCT01374503
	LY2624587	Lymphoma, leukemia	Phase I	[210] NCT01139788
	Hz515H7 (F50067)	Multiple myeloma, lymphoma, AML	Phase I	[211,213]
	MEDI3185	Multiple myeloma, Burkitt’s lymphoma, ovarian, lung	Preclinical	[212]
	Plerixafor (AMD3100, Mozobil^®^)	Breast, lung, CRC, prostate, pancreatic,	FDA approved for multiple myeloma and lymphoma	[214,215,216,217,218,219,220,221,222,223,224] NCT00694590, NCT05510544, NCT00903968, NCT00906945
	Mavorixafor (X4P-001)	RCC, melanoma, breast, Waldenström’s macroglobulinemia	Phase II	[225,226] NCT02823405, NCT05103917, NCT02667886, NCT02923531, NCT04274738
	USL311	Glioblastoma	Phase I/II	NCT02765165
	PRX177561	Glioblastoma	Preclinical	[227]
	MSX-122	Breast, HNSCC, uveal melanoma	Phase I	[228] NCT00591682
	Motixafortide (BL-8040, BKT-140, TNI4001)	AML, breast, pancreatic, solid tumors	Phase II	[218,228,229,230,231,232] NCT01838395, NCT02826486
	LY2510924	AML, solid tumors, RCC, SCLC	Phase II	[233,234,235] NCT02737072, NCT02652871, NCT01391130, NCT01439568
	CTCE-9908	Breast, solid tumors	Phase I/II	[236,237]
	IS4	Prostate, melanoma	Preclinical	[238]
CD133	Anti-CD133 mAb	CRC, breast	Preclinical	[256,257,258]
	BsAb-CIK	Pancreatic	Preclinical	[259]
	293C3-SDIE	CRC, leukemia	Preclinical	[260,261]
	Anti-CD133 CAR NK cells	Ovarian	Preclinical	[262]
	Anti-CD133 CAR T cells	SCLC, liver, CRC	Phase II	[263,264,265] NCT02541370, NCT02541370
	16x133 BiKE	CRC	Preclinical	[266]
	16x15x133 TriKE	CRC, breast, HNSCC, prostate, AML	Preclinical	[267]
	1615EpCAM133 TetraKE	CRC	Preclinical	[268]
	ICT-121	Glioblastoma	Phase I	[269] NCT02049489
	GMI	Lung	Preclinical	[270]
	dCD133KDEL	HNSCC, breast, ovarian	Preclinical	[271,272,273]
	AC133-saporin	CRC	Preclinical	[274]
	Celecoxib	CRC	FDA-approved NSAID for migraines	[275,276]
	Trifluridine	CRC	FDA-approved for metastatic CRC	[277]
	ACT001	Lung, glioma	Phase II	[278,279,280] ACTRN12616000228482, NCT05053880
	CD133 aptamers	HCC, breast	Preclinical	[281,282,284]
	LS-7	CRC, breast	Preclinical	[285]
	CRISPR/cas9	CRC	Preclinical	[286]
	siRNA	CRC	Preclinical	[287]
	AC133 NIR-PIT	Glioma	Preclinical	[288]
Nanog	IGT-PMO	Breast	Preclinical	[302]
	siRNA	CRC	Preclinical	[305]
	SAHA	HNSCC, lymphoma	FDA-approved for T-cell lymphoma	[307,308]
	PiB	Prostate	Preclinical	[309]
	Resveratrol	Glioblastoma	Preclinical	[313,314]
	Aspirin	CRC	FDA-approved NSAID	[315]
	Metformin	TNBC	FDA-approved for diabetes mellitus	[316]
	DFX, SP10	Esophageal	Preclinical	[317]
Notch	Crenigacestat (LY3039478)	Breast, CRC, lung, ovarian, glioblastoma, gastric, intrahepatic cholangiocarcinoma, multiple myeloma	Phase I	[327,328,329,330,331] NCT01695005, NCT02836600, NCT03502577
	LY900009	Solid tumors, lymphoma	Phase I	[333] NCT01158404
	Osugacestat (AL101, BMS-906024)	Breast, leukemia, adenoid cystic carcinoma, NSCLC	Phase II	[334,335,336,337,338] NCT03691207, NCT01653470
	RO4929097 (RG473)	Melanoma, CRC, sarcoma, pancreatic adenocarcinoma	Phase II	[340,344] NCT01116687, NCT01120275, NCT01154452, NCT01232829
	Nirogacestat (PF-03084014)	HCC, prostate, dermoid, breast	Phase III	[345,346,347,348,349] NCT01981551, NCT04195399, NCT03785964, NCT01876251
	DAPT	Osteosarcoma, gastric, adenoma	Preclinical	[350,351,352]
	MRK-0752	Breast, ovarian, solid tumors, PDAC, CNS malignancies	Phase II	[326,353,354,355,356,357] NCT01098344
	MRK-560	Leukemia	Preclinical	[358]
	Limantrafin (CB103)	Breast, leukemia	Preclinical	[359,360]
	ZLDI-8	NSCLC, CRC, HCC	Preclinical	[361,362,363]
	602.101	Breast	Preclinical	[364]
	Anti-Notch1 mAb	Lung, CRC	Preclinical	[365]
	Tarextumab (OMP-59R5)	Breast, lung, ovarian, pancreatic	Phase II	[366,367,368,369] NCT01277146, NCT01859741, NCT01647828
	Brontictuzumab (OMP-52M51)	Solid tumors, hematologic and lymphoid malignancies, CRC, ACC,	Phase I	[370,371] NCT01778439, NCT01778439
	PF-06650808	Breast	Phase I	[372] NCT02129205
	Enoticumab (REGN421)	Solid tumors	Phase I	[373] NCT00871559
	Demcizumab (OMP-21M18)	Ovarian, peritoneal, fallopian, NSCLC, pancreatic, solid tumors	Phase I	[374,375] NCT01952249, NCT01189968, NCT01189942, NCT02722954, NCT01952249
	MEDI0639	Solid tumors	Phase I	[376,377] NCT01952249
	MGZ01	Breast	Preclinical	[378]
	Navicixizumab (OMP-305B83)	Solid tumors, ovarian, peritoneal, fallopian	Phase I	[379,380] NCT02298387, NCT03030287
	ABT-165	Glioblastoma, CRC	Phase I	[381,382] NCT03368859
	Tarlatamab (AMG 757)	SCLC	Phase II	[383,384,385] NCT03319940, NCT05060016
	HPN328	SCLC, neuroendocrine	Phase I/II	[386] NCT04471727
	Rova-T	SCLC	Phase III	[387,388] NCT04471727
Wnt/β-catenin	Vantictumab (OMP-18R5)	CRC, breast, lung, pancreatic, solid tumors,	Phase I	[402,403,404,405] NCT01345201, NCT02005315,
	F2.A	PDAC	Preclinical	[406]
	IgG-2919	PDAC	Preclinical	[407]
	OTSA101	Synovial sarcoma	Phase I	[408,409,410,411] NCT01469975, NCT04176016
	Ipafricept (OMP-54F28)	Pancreatic, ovarian, solid tumors	Phase I	[413,414,415,416] NCT01608867, NCT02092363, NCT02050178
	Rosmantuzumab (OMP-131R10)	Leukemia, CRC	Phase I	[418,419] NCT02482441
	LGK974 (Wnt974)	HNSCC, CRC, solid tumors	Phase II	[420,421,422,423,424] NCT01351103, NCT0227813, NCT02649530
	ETC-159	CRC, pancreatic, solid tumors	Phase I	[425,426,427] NCT02521844, NCT02521844
	C59	Nasopharyngeal carcinoma	Preclinical	[428,429]
	RXC004	Pancreatic, CRC, solid tumors	Phase II	[430,431,432] NCT03447470, NCT04907851, NCT04907539
	XNW7201	Solid tumors	Phase I	NCT03901950
	CGX1321	GI tumors	Phase I	[433] NCT02675946
	Tegavivint (BC2059, tegatrabetan)	Desmoid, AML, multiple myeloma	Phase I/II	[434,435,436,437] NCT03459469, NCT04851119
	FH535	Breast, pancreatic, CRC	Preclinical	[400,438,439,440,441,442]
	Doxorubicin	Leukemia	Preclinical	[443]
	FOG-001	Solid tumors	Phase I	NCT05919264
	Carbamazepine	N/A	FDA-approved anti-epileptic	[444]
	FJ9	NSCLC	Preclinical	[445]
	dFz7-21	N/A	Preclinical	[446]
	Niclosamide	CRC, ovarian, prostate	FDA-approved anti-helminthic	[447,448,449] NCT02532114, NCT02519582
SOX2	ZF-522SKD/ ZF598SKD	Breast	Preclinical	[465]
	ATF/SOX2	Lung	Preclinical	[466]
	PIP-S2	N/A	Preclinical	[467]
	sP42	ESCC	Preclinical	[458]
	SOX2-iPEP	Breast, ovarian	Preclinical	[468]
	60030	Oligodendroglioma	Preclinical	[470]
	STEMVAC	Breast, NSCLC	Phase II	NCT05242965, NCT05455658, NCT02157051
	Rapamycin	Glioma	FDA-approved for perivascular epithelioid tumors	[473]
	MK2206	ESCC, breast	Phase II	[474,475] NCT01277757
	DC120	Nasopharyngeal carcinoma	Preclinical	[476]
	AZD4547	Lung	Phase II	[461,477] NCT01791985
	Gefitinib	NSCLC	FDA-approved for NSCLC	[478]
	Dasatinib	NSCLC	FDA-approved for CML	[478]
	LY294002	NSCLC	Preclinical	[478]
	Erlotinib	NSCLC	FDA-approved for NSCLC	[478]
	Gentian violet	Melanoma	FDA-approved antimycotic/antibacterial	[479]
	CBP30	Lung	Preclinical	[480]
	Pevonedistat (MLN4924)	Breast, NSCLC, leukemia, multiple myeloma, solid tumors, lymphoma	Phase III	[481,482,483] NCT03268954, NCT03323034, NCT03770260, NCT03965689
	APG-1387	Nasopharyngeal carcinoma	Preclinical	[462,484]
	FT234, FT895	NSCLC	Preclinical	[485]
	CBB1007	Lung	Preclinical	[486]
	Iadademstat (ORY-1001)	Breast, AML, SCLC	Phase II	[487,488] EUDRACT 2013-002447-29, NCT05420636, NCT05546580
Sonic Hedgehog	Vismodegib (GDC-0449)	Pancreatic, BCC, gastric, ovarian	FDA-approved for BCC	[496,497,498,499,500,501,502] NCT00607724, NCT00833417, NCT03052478, NCT00739661, NCT01088815
	Sonidegib (LDE225)	CML, BCC, NSCLC, SCLC, medulloblastoma, breast	FDA-approved for BCC	[503,504,505,506,507,508] NCT01327053, NCT01579929, NCT02027376, NCT01125800, NCT01456676
	Glasdegib (PF-04449913)	AML	FDA-approved for AML	[509,510,511,512] NCT03416179
	Cyclopamine	Breast	Preclinical	[513,514]
	IPI-269609	Pancreatic	Preclinical	[515]
	Saridegib (IPI-926)	Medulloblastoma, chondrosarcoma, HNSCC	Phase I	[495,516,517,518,519,520,521] NCT01255800, NCT01383538, NCT01310816
	BMS-833923 (XL139)	Esophageal, ovarian, AML, CML	Phase I	[522,523,524,525] NCT01218477
	Taladegib (LY2940680, ENV-101)	Medulloblastoma, BCC	Phase II	[526,527] NCT01919398, NCT02784795, NCT01226485, NCT05199584
	Hh003	CRC, pancreatic	Preclinical	[528]
	Itraconazole	Medulloblastoma, CRC, breast, biliary tract	FDA-approved anti-fungal	[529,530,531,532,533,534,535,536,537] NCT00769600
	ATO	SCLC, Ewing sarcoma, pancreatic	FDA-approved for acute promyelocytic leukemia	[538,539,540,541,542]
	GANT61	Melanoma, pancreatic, HCC	Preclinical	[492,494,543,544,545]
	MEDI-5304	CRC, pancreatic	Preclinical	[546]
	5E1	CRC, pancreatic, breast	Preclinical	[546,547,548,549]
	α-Ptch1	Pancreatic	Preclinical	[550]

## Figures and Tables

**Figure 1 ijms-25-04102-f001:**
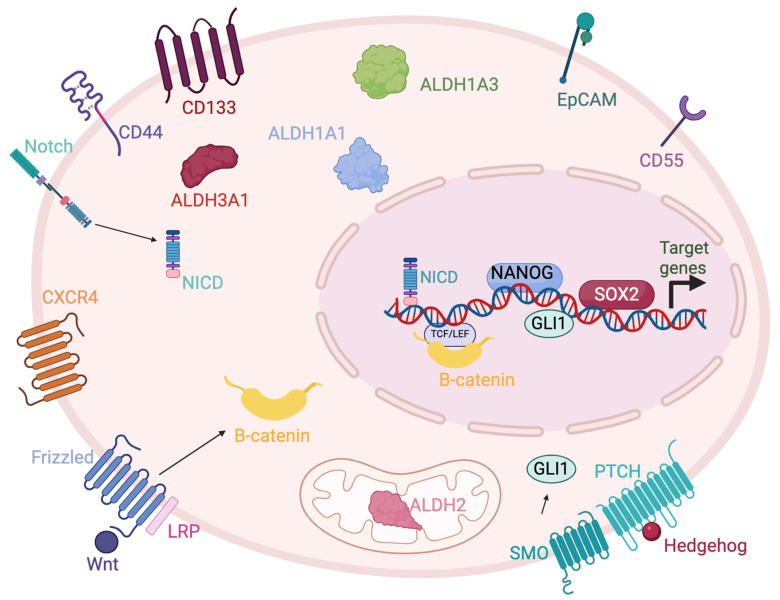
Cellular localization of CSC markers, pathways, and factors associated with CSCs in solid tumors as therapeutic targets. Schematic representation depicting the intracellular and membrane-bound distribution of key CSC markers, pathways, and factors associated with CSCs ALDH1A1, ALDH1A3, ALDH2, ALDH3A1, EpCAM, CD44, CD133, CD55, CXCR4, SOX2, Notch, Nanog, and SHH. Arrows depict signaling pathway intermediates.

**Figure 2 ijms-25-04102-f002:**
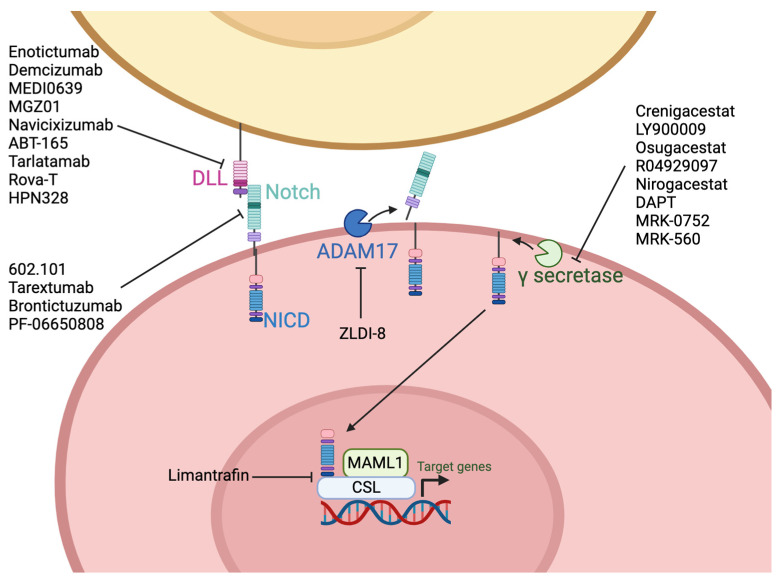
Notch signaling pathway and therapeutic interventions. Schematic representation depicting the therapeutic targets of the Notch signaling pathway. Delta-like ligand binding to the Notch receptor results in proteolytic cleavage of the Notch receptor, catalyzed by ADAM metalloproteases and γ-secretase to release the Notch intracellular domain. DLL: Delta-like ligand, NICD: Notch intracellular domain, ADAM17: A disintegrin and metalloproteinase 17, MAML: mastermind-like, CSL: CFB1/RBPJκ/Su(H)/LAG-1.

**Figure 3 ijms-25-04102-f003:**
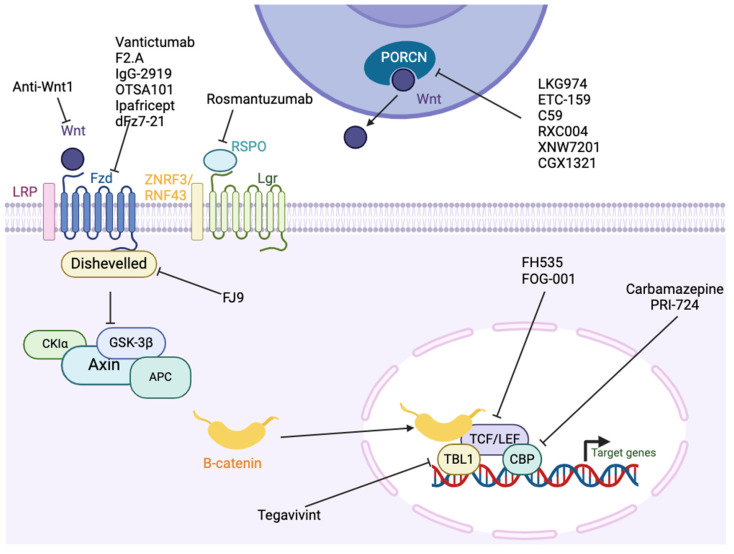
Therapeutic targets of the Wnt/β-catenin signaling pathway. Wnt ligands are activated by porcupine-mediated lipid modification and bind to Frizzled receptors and LRP co-receptors. R-spondin binding to Lgr receptors enhances Wnt signaling by sequestering ZNRF3 and RNF43 to stabilize the Wnt/Fzd interaction. Activation of the Fzd/LRP receptor complex leads to the recruitment and activation of Disheveled. In turn, Disheveled inhibits the β-catenin destruction complex. β-catenin translocates to the nucleus, where it displaces TBL1-containing corepressor complexes from TCF/LEF transcription factors to allow the transcription of target genes. CBP is recruited to the promoter regions of Wnt target genes. PORCN: Porcupine, Wnt: wingless-related integration site, LRP: low-density lipoprotein receptor-related protein, ZNRF3: zinc and ring finger 3, RNF43: ring finger protein 43, Lgr: leucine-rich repeat-containing G protein-coupled receptor, RSPO: R-spondin, GSK-3β: glycogen synthase kinase-3β, Axin: axis inhibition protein, APC: adenomatous polyposis coli, CK1α: casein kinase 1 α, TBL1: transducin beta-like 1, TCF/LEF: T-cell factor/lymphoid enhancer factor, CBP: CREB-binding protein.

**Figure 4 ijms-25-04102-f004:**
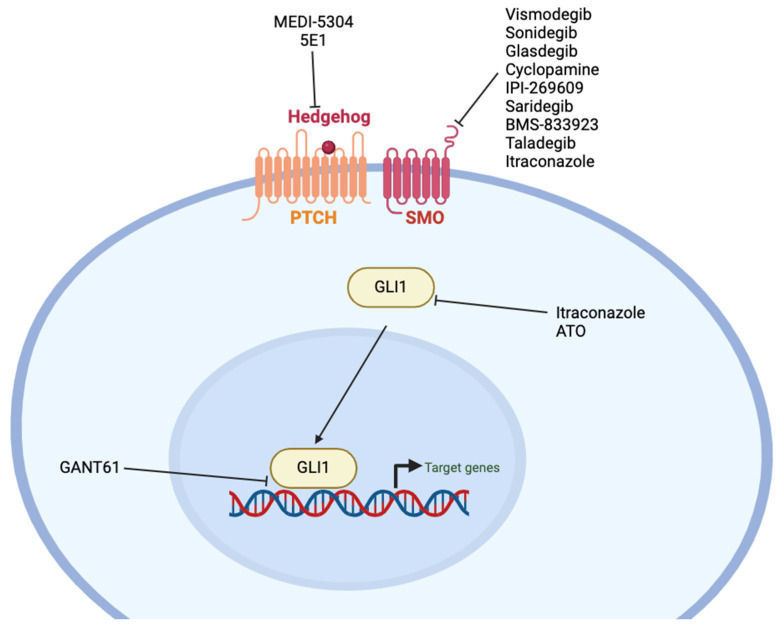
Therapeutic targets in the HH signaling pathway. Hedgehog ligands bind to Patched on the surface of target cells, relieving Patched-mediated inhibition of Smoothened. Activated Smoothened leads to activation of the Gli family of transcription factor, which promotes the expression of target genes. PTCH: Patched, SMO: smoothened, GLI1: Glioma-associated oncogene homolog 1.

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
