# Peer review of "Informed by Cancer Stem Cells of Solid Tumors: Advances in Treatments Targeting Tumor-Promoting Factors and Pathways"

_ijms, 2024, doi:10.3390/ijms25074102_

Round 1

Reviewer 1 Report

Comments and Suggestions for Authors

This manuscript deals with the cancer stem cells. The authors have considered several potential and effective target molecules for therapeutic intervention.

I would say this manuscript is really comprehensive and up.to-date.

On the other hand, it is not focused on a specific CSC. However, the CSC in haematological malignancies are not considered at all.

Actually, the authors considered mainly solid tumors. This should be indicated in the title. 

Also, it would be of interest in a specific paragraph to point out how much the target molecules of CSC are indeed specific for CSC. Several of the molecules listed are expressed by tumor cells that are not CSC as well as healthy cells. 

Comments on the Quality of English Language

English is good.

Author Response

We thank the reviewers for their positive feedback and helpful comments for improving the manuscript. Below we describe how we have addressed each suggestion.

Reviewer 1

This manuscript deals with the cancer stem cells. The authors have considered several potential and effective target molecules for therapeutic intervention.

I would say this manuscript is really comprehensive and up.to-date.

On the other hand, it is not focused on a specific CSC. However, the CSC in hematological malignancies are not considered at all.

Actually, the authors considered mainly solid tumors. This should be indicated in the title. 

RESPONSE: Yes, we have modified the title as suggested and we mention this in the Abstract, the Figure 1 caption, and in the Introduction, please see lines 53-55.

Also, it would be of interest in a specific paragraph to point out how much the target molecules of CSC are indeed specific for CSC. Several of the molecules listed are expressed by tumor cells that are not CSC as well as healthy cells. 

RESPONSE: Thank you for the suggestion, we have added the paragraph in the Introduction as suggested. Please see lines 61-71.

Reviewer 2 Report

Comments and Suggestions for Authors

Informed by cancer stem cells: advances in treatments targeting tumor-promoting factors and pathways 

My first comment is that this is not an article, it’s a book/ book chapter. I’m not sure if there are any regulations and rules at IJMS for article length but this is unusually long for an article. Have the authors considered publishing this as part of a book?

The abstract highlights some of the core properties of CSCs and as stated by the authors, it is indeed comprehensive.

Markers such as ALDH, CD44, and CD133 and signalling pathways such as Wnt, Notch and Hh are mentioned.

The authors aim to link various therapeutic modalities with CSCs and extend this to clinical trials. The conclusion states that advances made in the CSC field, have impacted effective and durable responses.

Overall, the abstract is relevant and informative, thanks.

The authors mention there is only a low number of CSCs in tumour volumes, they could also refer to the melanoma experiment in which larger percentages of CSC were identified.  I suspect the 25% cited refers to that.

The authors could try to distinguish between CSCs, cancer-initiating cells and persister cells since these are not the same.

The authors mention that CSCs have specific markers, kindly give some information about what markers CSCs show in each cancer.

Figure 1 is useful and informative. The authors could mention if this schematic represents representative CSCs.

The authors then move on to ALDH and useful information is provided about this marker and its subtypes are then focused on.

In 2.1. the authors seem to be focusing on chemotherapy resistance-related topics and also migration. Since the subsection does not give specific details about what the focus will be, it might be an idea to make this clear. I can also see the topics introduced in 2.2. are also linked to chemoresistance. Therefore, giving a specific header about the link between these isoforms and treatment responses would be useful. 

Section 2.5 is interesting and IC50s for this inhibitor are mentioned and this compound is examined across some cancer types. Other inhibitors are explained as well. 

Then the authors move on to EpCAM and antibodies targeting this marker are then mentioned. I would like to suggest the same “informative subheading title” for these subsections. Please apply this to the rest of the titles. Interestingly, clinical trials targeting this marker are also introduced.  

I realise that towards the end of the manuscript, the authors provided a massive table with a summary of each of the main links given for topics 1-12 (Aldh- Hh). But despite this, the manuscript is text-heavy and a better balance of figures would be ideal. So I would suggest that the authors provide a summary figure/scheme (of their choice) for each of these 12 subsections. These could even be flowcharts or any smart-art schemes (in PowerPoint) that could help summarise the topics of each of these 12 subsections.  

As the authors mention, CD44 is the most important marker of the CSC population and they do refer to what cancers express this marker on their CSC surface. Of course, the Al-Hajj BC studies are cited, which is great. Again, various treatment modalities are offered such as antibodies, NIR-PIT and pharmacological inhibitors.

5. CD55 is an interesting example of a marker since it is quite unusual and non-canonical and seems to be present on BC CSCs. It seems that this marker is also linked to chemoresistance. 

It is also interesting that not all treatment modules are available for all markers, such as CAR-T cells and NK-mediated cell cytotoxicity for CD133. Or immunotoxins for the same section.

Also, for Nanog, it is mainly downregulation and pharmacological inhibition rather than other types of targeting perhaps because it is a transcription factor. The sections about the transcription factors could have a separate heading or introduction to outline if manipulating a transcription factor is different to a surface/ secreted-secreted marker. 

The scientific content of this study is sound, the study is very comprehensive and the points raised above can assist in making the content more readable. Overall, I commend the authors for such a coherent work. 

Author Response

We thank the Reviewer for their positive feedback and helpful comments to improve the manuscript. We have incorporated the suggestions as noted below.

Reviewer 2

My first comment is that this is not an article, it’s a book/ book chapter. I’m not sure if there are any regulations and rules at IJMS for article length but this is unusually long for an article. Have the authors considered publishing this as part of a book?

The abstract highlights some of the core properties of CSCs and as stated by the authors, it is indeed comprehensive.

Markers such as ALDH, CD44, and CD133 and signalling pathways such as Wnt, Notch and Hh are mentioned.

The authors aim to link various therapeutic modalities with CSCs and extend this to clinical trials. The conclusion states that advances made in the CSC field, have impacted effective and durable responses.

Overall, the abstract is relevant and informative, thanks.

The authors mention there is only a low number of CSCs in tumour volumes, they could also refer to the melanoma experiment in which larger percentages of CSC were identified.  I suspect the 25% cited refers to that.

The authors could try to distinguish between CSCs, cancer-initiating cells and persister cells since these are not the same.

RESPONSE: Thank you for the suggestion, we have this distinction to the Introduction as suggested. Please see lines 71-75.

The authors mention that CSCs have specific markers, kindly give some information about what markers CSCs show in each cancer.

RESPONSE: This information is provided in most of the individual sections; however, it was missing for ALDH1A3 and CD55. This has now been added. Please see lines 137-138 and 567.

Figure 1 is useful and informative. The authors could mention if this schematic represents representative CSCs.

RESPONSE: Thank you, we have mentioned this now, please see lines 59-60. 

The authors then move on to ALDH and useful information is provided about this marker and its subtypes are then focused on.

In 2.1. the authors seem to be focusing on chemotherapy resistance-related topics and also migration. Since the subsection does not give specific details about what the focus will be, it might be an idea to make this clear. I can also see the topics introduced in 2.2. are also linked to chemoresistance. Therefore, giving a specific header about the link between these isoforms and treatment responses would be useful. 

RESPONSE: We have tried to provide more explanation for the subsections and what is described in each subsection. Please see lines 102-105.

Section 2.5 is interesting and IC50s for this inhibitor are mentioned and this compound is examined across some cancer types. Other inhibitors are explained as well. 

Then the authors move on to EpCAM and antibodies targeting this marker are then mentioned. I would like to suggest the same “informative subheading title” for these subsections. Please apply this to the rest of the titles. Interestingly, clinical trials targeting this marker are also introduced.  

I realise that towards the end of the manuscript, the authors provided a massive table with a summary of each of the main links given for topics 1-12 (Aldh- Hh). But despite this, the manuscript is text-heavy and a better balance of figures would be ideal. So I would suggest that the authors provide a summary figure/scheme (of their choice) for each of these 12 subsections. These could even be flowcharts or any smart-art schemes (in PowerPoint) that could help summarise the topics of each of these 12 subsections. 

RESPONSE: This is a good suggestion; however, we couldn’t think of good figures for each of the sections that would add information, since most of the sections are just targeting one molecule that is already shown in Fig. 1.  However, we have added three additional new figures for the pathways of Nanog, Wnt, and Hedgehog since these involve multiple players that we agree could benefit from additional illustrative summary. Please see new Figures 2, 3 and 4.

As the authors mention, CD44 is the most important marker of the CSC population and they do refer to what cancers express this marker on their CSC surface. Of course, the Al-Hajj BC studies are cited, which is great. Again, various treatment modalities are offered such as antibodies, NIR-PIT and pharmacological inhibitors.

  1. CD55 is an interesting example of a marker since it is quite unusual and non-canonical and seems to be present on BC CSCs. It seems that this marker is also linked to chemoresistance. 

It is also interesting that not all treatment modules are available for all markers, such as CAR-T cells and NK-mediated cell cytotoxicity for CD133. Or immunotoxins for the same section.

Also, for Nanog, it is mainly downregulation and pharmacological inhibition rather than other types of targeting perhaps because it is a transcription factor. The sections about the transcription factors could have a separate heading or introduction to outline if manipulating a transcription factor is different to a surface/ secreted-secreted marker. 

RESPONSE: We are not sure how to best address this suggestion and have therefore elected to not make the suggested change. We hope this is ok.

The scientific content of this study is sound, the study is very comprehensive and the points raised above can assist in making the content more readable. Overall, I commend the authors for such a coherent work. 

Round 2

Reviewer 2 Report

Comments and Suggestions for Authors

I have no further comments.